# Lightweight, Pre-trained Transformers for Remote Sensing Timeseries

## Abstract

Machine learning models for parsing remote sensing data have a wide range of societally relevant applications, but labels used to train these models can be difficult or impossible to acquire. This challenge has spurred research into self-supervised learning for remote sensing data aiming to unlock the use of machine learning in geographies or application domains where labelled datasets are small. Current self-supervised learning approaches for remote sensing data draw significant inspiration from techniques applied to natural images. However, remote sensing data has important differences from natural images – for example, the temporal dimension is critical for many tasks and data is collected from many complementary sensors. We show we can create significantly smaller performant models by designing architectures and self-supervised training techniques specifically for remote sensing data. We introduce the **P**retrained **Re**mote **S**ensing **T**ransf**o**rmer (Presto), a transformer-based model pre-trained on remote sensing pixel-timeseries data. Presto excels at a wide variety of globally distributed remote sensing tasks and performs competitively with much larger models while requiring far less compute. Presto can be used for transfer learning or as a feature extractor for simple models, enabling efficient deployment at scale.

## 1 Introduction & Related Work

Recent advances in machine learning capabilities combined with vast remote sensing datasets have provided many societally beneficial outcomes ranging from tracking progress on sustainable development goals (Ferreira et al., 2020) to improved weather forecasting (English et al., 2013; Voosen, 2020) to disaster management (Kansakar & Hossain, 2016). However, the remote sensing data modality has several characteristics that are important to consider when designing machine learning algorithms in this domain:

- **Highly multi-modal data**: According to The Union of Concerned Scientists (2023), there were 1,142 Earth-observing satellites orbiting Earth in January 2023. These satellites carry a wide range of sensors, including synthetic aperture radar (Torres et al., 2012) and multispectral optical sensors (Drusch et al., 2012). In addition, there are many derived data products which are created by the manipulation of these raw data sources (such as digital elevation maps (Rabus et al., 2003)). Rao et al. (2020) found that combining a variety of direct observations and derived products yielded the strongest fuel moisture prediction performance.
- **A highly informative temporal dimension**: The Earth's highly dynamic nature (Yifang et al., 2015) and the relatively coarse resolution of freely available satellite data means that in remote sensing, the temporal dimension is critical for many downstream tasks (Rußwurm et al., 2023). A common approach by remote sensing practitioners is therefore to train single pixel-timeseries models (Rußwurm et al., 2023; Sainte Fare Garnot et al., 2020; Pelletier et al., 2019; Wang et al., 2020; Hengl et al., 2017) (as opposed to models which instead emphasize the spatial dimension, which is the case in natural imagery).
- **Unique metadata**: All remote sensing data is associated with (i) the location data describing the place being imaged, and (ii) the timestamp at which the image was captured. This information can be useful for machine learning algorithms. For example, You et al. (2017) found that supplementing a yield-estimation model with time and location data significantly improved performance.

Datasets for remote sensing often have very few labels (Helber et al., 2019), which may be unreliable (Bressan et al., 2022; Yifang et al., 2015) or absent, especially for under-resourced geographies (Kerner et al., 2020; Nakalembe et al., 2021), leading to poor global generalization (Yifang et al., 2015). This limited availability of labels combined with plentiful unlabeled data has spurred the investigation of self-supervised learning algorithms for remote sensing data (Reed et al., 2022; Cong et al., 2022; Jean et al., 2019; Manas et al., 2021; Ayush et al., 2021).

Previous approaches investigating self-supervised learning for remote sensing primarily treat remote sensing data as analogous to natural imagery, and therefore attempt to co-opt methods and architectures originally designed for natural imagery (i.e., ground-level photography) – for example, by using a ResNet (He et al., 2016) backbone (Jean et al., 2019; Manas et al., 2021; Ayush et al., 2021), or by adapting masked autoencoding for image classification (He et al., 2022) to satellite imagery (Reed et al., 2022; Cong et al., 2022). These models fail to leverage all the attributes of remote sensing data (for example, models which can only ingest data from a single RGB sensor, or which do not consider the temporal dimension of the data).

Transformers have been investigated for remote sensing timeseries, either as unmodified architectures (Rußwurm & Körner, 2020) or as architectures designed for specific tasks (Sainte Fare Garnot et al., 2020; Tarasiou et al., 2023). While several pre-training methods have been proposed for transformers with remote sensing timeseries (Yuan & Lin, 2020; Yuan et al., 2022; 2023), these have not aimed at multi-task, global applicability, having been pre-trained and evaluated on highly local areas (e.g., central California) and evaluated only for a single task (e.g., crop type classification).

To take advantage of the unique characteristics, global scope, and broad applicability of remote sensing data, we introduce the **P**retrained **Re**mote **S**ensing **T**ransf**o**rmer (Presto), a lightweight transformer-based model designed to ingest pixel-timeseries inputs from a variety of Earth observation sensors and data products. We tailor the self-supervised learning process to learn from multiple data sources and from the temporal dimension of the data so that Presto learns powerful representations of remote sensing data. These representations can be efficiently adapted to a wide range of globally distributed sensing tasks. Presto is also robust to missing input sensors or timesteps, excelling even in image-based tasks where the temporal dimension is completely absent.

When deployed, models built for remote sensing data are typically used to make contiguous geospatial predictions over millions (or billions) of samples to form a predicted map. The computational performance of models is therefore one of the primary considerations at deployment time: Van Tricht (2021), Hengl et al. (2017) and Robinson et al. (2019) all prioritized efficiency over accuracy when deploying remote sensing models at scale. This has limited the adoption of large models with ViT or ResNet backbones for large scale mapping. In comparison, Presto is highly accurate despite having $1000\times$ fewer trainable parameters than ViT- or ResNet-based models (and requiring orders of magnitudes fewer FLOPs at inference time), making it especially well-suited to real-world deployment.

In summary, the main contributions of this work are:

- We introduce Presto, a lightweight transformer-based model designed for Earth observation data from a potentially diverse set of sources or sensors. We pre-train Presto with a novel MAE-based self-supervised methodology designed to leverage the structure in multi-sensor pixel-timeseries.
- We show that Presto is competitive with state-of-the-art models across a variety of geographies, dataset sizes, and task types while often requiring significantly less compute. Although Presto only ingests pixel timeseries, it is also performant on tasks where the spatial component is important.
- We show that Presto is competitive even on tasks where a subset of pre-training channels or timesteps are available (including tasks with only a single time step and/or a single sensory modality), showing that leveraging multi-sensor timeseries during the pre-training process confers a significant advantage in the model's learned representations.

## 2 METHOD

We aim to learn a model, $f$, which can learn useful representations in a self-supervised manner given unlabelled remote sensing pixel-timeseries data, $x$. This model can then be applied to a wide variety of downstream remote sensing tasks. Importantly, while we leverage the structure of multi-sensor pixel-timeseries data for self-supervised pre-training, the model can then be fine-tuned on data with a different number of timesteps or sensors than the pre-training input format.

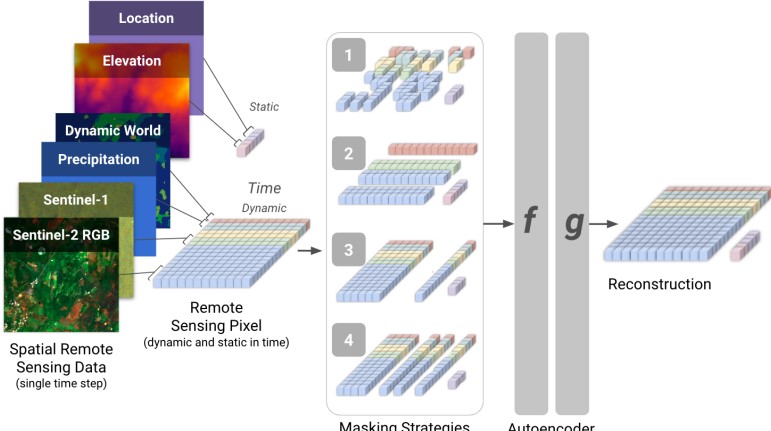

Figure 1: **Presto learns from structurally-masked remote sensing pixel-timeseries**. Specifically, we construct a multi-sensor remote sensing pixel-timeseries, and randomly select one of the four masking strategies described in Section 2.3. The encoder-decoder model is then trained to reconstruct the original timeseries. At fine-tuning time, we discard the decoder and only use the encoder's output. The downstream task may have incomplete inputs (missing timesteps or sensors) since the encoder is specifically trained on such inputs. Presto receives both static-in-time and dynamic-in-time inputs and the location metadata of each pixel timeseries. In this figure, a cube represents a pixel. The 2D arrangement is for clarity and does not represent concatenation of the pixels.

Our approach is based on the masked autoencoding framework (He et al., 2022), in which the network architecture includes both an encoder ($f$) and a decoder ($g$). During pre-training, part of the input is masked out and the encoder embeds the remaining (non-masked) part of the input. The decoder aims to reconstruct the masked-out part of the input, given the encoder's output. At fine-tuning time, we discard $g$ and only use $f$ (either as a feature extractor or a fine-tuneable model) for downstream tasks. In the sections below, we discuss how Presto adapts this general framework for multi-sensor remote sensing timeseries data. An overview of the Presto pre-training methodology is shown in Figure 1, and full pre-training details are in Section A.1.

## 2.1 PRE-TRAINING DATA

Remote sensing models can be deployed in a wide range of geographies, with few labelled datapoints available at fine-tuning time (Kerner et al., 2020; Böhm et al., 2022). We therefore aim to collect a globally representative pre-training dataset. We followed the sampling strategy of Dynamic World (Brown et al., 2022) to construct a dataset of 21.5M pixel samples, each having 10m per pixel resolution. Appendix A.1.1 describes the pre-training dataset construction process in detail. Presto is trained on pixel-timeseries of 12-month contiguous intervals, sampled from a 2-year period from the beginning of 2020 until the end of 2021, with each month represented by one timestep (similar to the approach adopted in Tseng et al. (2021)). Derived data products which result from analysis of lower level data (e.g., (Parkinson et al., 2006)) can significantly improve model performance (Rao et al., 2020; Hengl et al., 2017). We therefore pre-train Presto on a diverse range of directly-sensed and derived Earth observation products, exported using Google Earth Engine (Gorelick et al., 2017).

A pre-training batch contains a number of pixel-timeseries, each of which consists of a concatenation of dynamic-in-time datapoints with each timestep representing a month (yielding $T = 12$ timesteps in total). To every pixel-timeseries we append a number of static-in-time features. We leverage the following data products when pretraining Presto:

- **Sentinel-1 Synthetic Aperture Radar observations** (S1): The VV (emit and receive at vertical polarization) and VH (emit at vertical and receive at horizontal polarization) bands: 2 real-valued dynamic values per monthly timestep.
- **Sentinel-2 Multispectral images** (S2): We removed the 60m resolution bands, yielding bands with 10m and 20m resolution with channels in the visible, near-infrared and short-wave infrared range: 10 real-valued dynamic values per timestep.

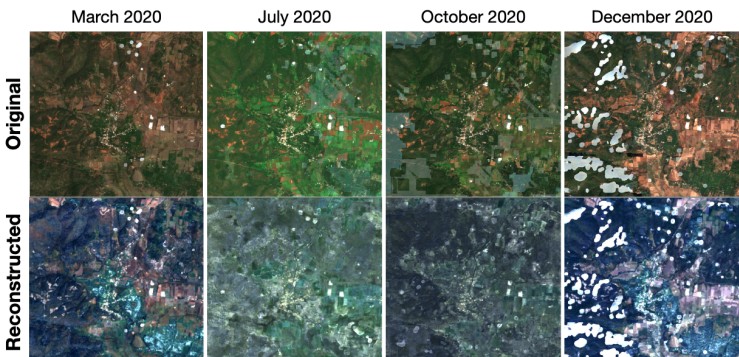

Figure 2: **Presto learns to reconstruct channels that are completely masked in a spatially cohesive manner**. In this experiment, we mask the Sentinel-2 RGB channels (all other channels are left unmasked); Presto is able to reconstruct these channels even when they are absent from the input. The reconstructions are spatially consistent even though Presto only receives single pixel inputs.

- **ERA5 Climate Reanalysis Meteorological data** (ERA5): Monthly total precipitation and temperature at 2 metres above the ground: 2 real-valued dynamic values per timestep.
- **NDVI** (Rouse et al., 1974): Computed from the red (B4) and near-infrared (B8) Sentinel-2 bands: 1 real-valued dynamic value per timestep.
- **Dynamic World Land Cover classes** (DW, Brown et al., 2022): Land cover classes produced for every non-cloudy Sentinel-2 image: 1 dynamic categorical value from the set of possible classes $\mathcal{V}$ per timestep. We took the mode of classes for all timesteps within a month.
- **Topography data** (TG), from the Shuttle Radar Topography Mission's Digital Elevation Model: The elevation and slope of each pixel, real-valued and static in time.
- **Coordinates** (Loc): 3D static in time Cartesian coordinates computed from the latitude and longitude of the pixel's geographical location: $s_{\mathrm{Loc}} = [\cos(\mathrm{lat}) \times \cos(\mathrm{lon}), \cos(\mathrm{lat}) \times \sin(\mathrm{lon}), \sin(\mathrm{lat})]$.

Hence, one pre-training sample $x$, comprising a pixel-timeseries $t \in [\mathbb{R}^{T \times 15}; \mathcal{V}^{T \times 1}]$ and static variables $s \in \mathbb{R}^{1 \times 5}$, is summarized as follows. From now on, we use "pixel-timeseries" to refer to both the dynamic and the static variables.

$$x = \left[ \{t_i^{\mathrm{S1}};\ t_i^{\mathrm{S2}};\ t_i^{\mathrm{ERA5}};\ t_i^{\mathrm{NDVI}};\ t_i^{\mathrm{DW}} \mid i = 1, ..., 12\};\ s^{\mathrm{TG}};\ s^{\mathrm{Loc}} \right] \tag{1}$$

## 2.2 Encoding and Tokenization

The pixel-timeseries $x$ is transformed into a number of tokens (each represented by an embedding $e$) to be processed by the Presto transformer. Per timestep $0 \leq i < T$, the input variables are split into channel groups $\mathcal{C}$ according to their type of sensor or source: e.g., the S1 bands form one channel group (we describe these groups in more detail in Appendix A.1.2). Each real-valued channel group represents a different sensor, native spatial resolution or (in the case of Sentinel-2 channel-groups) region of the electromagnetic spectrum. We therefore projected them to a common latent space of dimension $d_e$ by separate learned linear projections $h$: e.g., $e_i^{\mathrm{S1}} = h^{\mathrm{S1}}(t_i^{\mathrm{S1}})$. The Dynamic World classes are categorical, so they are embedded by indexing into an embedding matrix.

Unlike natural images, in which the data and its label are self-contained, remote sensing labels are inherently associated to a place and time on Earth (i.e., a latitude/longitude and timestamp). In addition, while each natural image contains the same RGB channels, Presto's pixel-timeseries represent channels from different remote sensing data products. We therefore want to communicate to the model: (i) the location of the datapoint (already present in the input as static variable through coordinates $s_{\mathrm{Loc}}$), and a variable's (ii) timestamp and (iii) channel group. We do this by adding encodings to the previously described embeddings $e$. The complete encoding is a concatenation of the positional, month and learned channel encodings and has dimension $d_e$.

- **Positional** We use the sinusoidal positional encoding $p_{\mathrm{sin}}$ originally used by transformer models (Vaswani et al., 2017), where variables from the same timestep get the same encoding.

| Channel Groups | Random | Timesteps | Contiguous Timesteps | F1 Score |
|:---:|:---:|:---:|:---:|:---:|
| ✓ | | | | 0.646 |
| | ✓ | | | 0.653 |
| | | ✓ | | 0.664 |
| | | | ✓ | 0.649 |
| ✓ | ✓ | ✓ | ✓ | **0.665** |

Table 1: **Structured masking strategies yield the best downstream performance**. We measured $\text{Presto}_R$'s F1 score on the CropHarvest validation task. Combining structured strategies outperformed the "Random" masking employed by He et al. (2022).

- **Month** We add an encoding $p_{\text{month}}$ that represents the month being captured by each token. This is done because we expect timesteps from similar months to have similar features even if they are from different years. We assign an integer to each month ranging from 0 to 11, yielding:

$$p_{\text{month},2i} = \sin\left((2\pi \times \text{month})/12\right) \qquad p_{\text{month},2i+1} = \cos\left((2\pi \times \text{month})/12\right) \qquad (2)$$

For static-in-time variables, the positional and month encodings are set to zero.

- **Channel Group** Finally, each token is associated with a set of input channels. In multi-spectral SatMAE (Cong et al., 2022), a fixed encoding is used to communicate input-band information, which is possible because only input data from one sensor (Sentinel-2) is used, with different channels representing different wavelengths. However, since Presto uses a number of different remote sensing products, we apply a learnable encoding $p_{\text{channel}}$ for each channel group from the set of possible channel groups $\mathcal{C} = \{\text{S1}, \text{S2 RGB}, ..., \text{ERA5}, \text{TG}, \text{Loc}\}$.

The transformer input $E \in \mathbb{R}^{(T \cdot |\mathcal{C}_{\text{dynamic}}| + |\mathcal{C}_{\text{static}}|) \times d_e}$ (for encoder dimension $d_e$) is a concatenation of:

- Dynamic variables: $e_i^c = h^c(t_i^c) + [p_{\text{channel}}^c;\ p_{\sin(i)};\ p_{\text{month}(i)}]$, for timesteps $i$ and channel groups $c$
- Topological data: $e^{\text{TG}} = h^{\text{TG}}(s^{\text{TG}}) + [p_{\text{channel}}^{\text{TG}};\ 0;\ 0]$
- Coordinates: $e^{\text{Loc}} = h^{\text{Loc}}(s^{\text{Loc}})$

## 2.3 Pre-training via Structured Masking

A goal of Presto is to perform well even with incomplete inputs (i.e., when there are fewer or missing timesteps, when channels are missing, or both). When masking out part of the input $x$, we therefore tailor the masking strategies to encourage the model to learn representations that perform well specifically when given a subset of bands or timesteps for downstream tasks. For a $T \times D$ input of $T$ timesteps and $D$ total input channels, we use the following masking techniques (illustrated in Figure 1), where Presto considers a token to be a $1 \times d$ input (a single timestep of $d$ grouped channels). The coordinates are never masked (but the static topological tokens can be).

1. **Random**: $(t \times d)$ masked values, with $t < T$ and $d < D$
2. **Channel-groups**: $(T \times d)$ masked values, with $d < D$
3. **Contiguous timesteps**: $(t \times D)$ masked values, with $t < T$
4. **Timesteps**: $(t \times D)$ masked values, with $t < T$

For each instance, we randomly sample from the above strategies to construct a mask. Table 1 shows results from ablating the masking strategies, measured against a validation task (classifying CropHarvest datapoints - excluding the evaluation data described in Section 3.1 - according to their FAO indiciative crop classifications). Unlike other masked-autoencoder methods (Cong et al., 2022; He et al., 2022), we find that combining structured masking strategies outperforms random masking. In addition, we find that Presto can reconstruct entirely missing channel groups in Figure 2.

Since the masked-autoencoder model is trained on both categorical and continuous timeseries inputs, the model's loss must also be calculated using both categorical and continuous reconstructions. This is achieved using the following loss function, which balances the loss for every batch so that each reconstructed value receives the same weighting in the final loss: $\mathcal{L}_{\text{total}} = \mathcal{L}_{\text{MSE}} + \lambda \frac{N_{\text{cat}}}{N_{\text{cont}}} \mathcal{L}_{\text{CE}}$. $\mathcal{L}_{\text{MSE}}$ is the mean squared error reconstruction loss used for the continuous values, $\mathcal{L}_{\text{CE}}$ is the cross entropy loss used for the categorical values, $N_{\text{cont}}$ is the number of masked continuous values and $N_{\text{cat}}$ is the number of masked categorical values (in the batch). $\lambda$ is a hyperparameter, which we set to 2.

Table 2: **We evaluate Presto on a wide variety of downstream tasks**. There is diversity in terms of input-data composition (numbers of timesteps and channels), geographic area and training size.

| Dataset | Task | Region | Timesteps | Channels | Training size |
|---|---|---|---|---|---|
| CropHarvest | Segmentation | Kenya
Brazil
Togo | 12 | 19 | 1,345
203
1,319 |
| TreeSat | Multi-Label Classification | Germany | 1 | 2 (S1)
11 (S2) | 45,337 |
| EuroSat | Classification | Europe | 1 | 3 (RGB)
11 (MS) | 21,600 |
| Fuel Moisture | Regression | USA | 3 | 19 | 1,578 |
| Algae Blooms | Regression | USA | 12 | 19 | 777 |

Table 3: **Adapting Presto to downstream tasks is computationally efficient**. We show F1 scores on the CropHarvest tasks. TIML and MOSAIKS-1D do not receive Dynamic World as input, so we evaluated Presto with and without it for fair comparison. In both cases, Presto outperforms these models while requiring the adaptation of far fewer parameters.

| | Number of parameters | | | | | |
| Model | Total | Finetuned | Kenya | Brazil | Togo | Mean |
|---|---|---|---|---|---|---|
| Random Forest | | | 0.559 | 0.000 | 0.756 | 0.441 |
| MOSAIKS-1D$_R$ | 418K | 8193 | 0.790 | 0.746 | 0.679 | 0.738 |
| TIML | 91K | 91K | 0.838 | 0.835 | 0.732 | 0.802 |
| Presto$_R$ | 401K | 129 | 0.816 | **0.891** | **0.798** | 0.835 |
| no DW | | | **0.861** | 0.888 | 0.760 | **0.836** |

## 3 EVALUATION

We aim to demonstrate the utility of Presto for a diversity of tasks, geographic task-locations, input data modalities and downstream dataset sizes. This diversity is demonstrated in Table 2. For downstream evaluation, we take the encoder-decoder model learned during pre-training and discard the decoder. As in (He et al., 2022), we pass a global-pool of all the encoder's output tokens to a downstream classifier. We evaluate the performance of three different models (Presto$_R$, Presto$_{RF}$, Presto$_{FT}$, defined below) built on top of Presto's encoder:

- **Feature extraction.** Rolf et al. (2021) demonstrated the utility of neural networks as feature-extractors on top of which computationally efficient classifiers could be trained. Presto$_R$ and Presto$_{RF}$ consist respectively of linear or logistic regressions and random forests trained on Presto's embeddings. Since only the regression/random forest is trained, not the Presto encoder, this a computationally efficient method for adapting Presto to a wide range of tasks.
- **Fine-tuning**. Presto$_{FT}$ consists of the Presto encoder, followed by a linear transform of the pooled tokens to the desired outputs. This entire model (the encoder and the linear transformation) is fine-tuned on the evaluation tasks using fixed hyperparameters. We used a subset of the training data for validation and early stopping.

We compare Presto to a number of self-supervised pre-training baselines, as well as state-of-the-art task specific baselines (Section A.3). We adapt MOSAIKS (Rolf et al., 2021) for timeseries data by convolving over the temporal dimension instead of the spatial dimension. We use the output features with random forests (MOSAIKS-1D$_{RF}$) and with regressions (MOSAIKS-1D$_R$).

### 3.1 TIMESERIES TASKS

- **Crop type Segmentation**: The CropHarvest (Tseng et al., 2021) evaluation datasets consist of classifying pixels as (i) maize in Kenya, (ii) coffee in Brazil and (iii) cropland in Togo. We

| Model | Data | Weighted | | Micro | |
|---|---|---|---|---|---|
| | | $F_1$ | mAP | $F_1$ | mAP |
| MLP | | 10.09 | 29.42 | 12.82 | 33.09 |
| LightGBM | S1 | 11.86 | 32.79 | 14.07 | 35.11 |
| Presto$_{RF}$ | | **19.79** | **35.76** | **22.92** | **38.69** |
| MLP | | **51.97** | **64.19** | **54.59** | **65.83** |
| LightGBM | S2 | 48.17 | 61.99 | 52.52 | 61.66 |
| Presto$_{RF}$ | | 46.26 | 60.88 | 50.41 | 63.24 |

Table 4: Results on the TreeSatAI dataset. We compare Presto to the dataset's benchmark models. The MLPs contain 3 layers (with 563K-723K parameters respectively) and are tuned for this task, whereas we freeze the Presto encoder's 401k parameters and train a random forest on its outputs with default scikit-learn hyperparameters.

compare Presto to the baselines provided by CropHarvest and to Task-Informed Meta-Learning (TIML, Tseng et al., 2022), a method that achieved state-of-the-art results on these datasets.

- **Fuel Moisture**: The live fuel moisture dataset (Rao et al., 2020) measures live fuel moisture content in the Western U.S. Rao et al. (2020) baseline used 5-fold cross validation to evaluate model performance; for future comparability, we use a geographically partitioned test set.
- **Algae Blooms**: The algae blooms dataset (alg, 2023) measures the severity of cyanobacterial algal blooms in different parts of the U.S. (we use the subset in the Midwestern U.S.). The dataset was originally released as part of a competition, so the test data is not available. In addition, competitors could download a range of Earth observation datasets to train their models, making direct comparisons to competition results difficult. We benchmark against a regression and a random forest (since the winning solution used a tree-based method), and use a geographically partitioned test set.

## 3.2 IMAGE-BASED TASKS

Presto is designed to ingest pixel-timeseries (and not single timestep images, as is the case for image-based tasks). However, many remote sensing benchmarks consist of static-in-time imagery, which represents an out-of-domain evaluation relative to Presto's pretraining. When a single prediction is required for an entire image, we use the following approach to obtain per-image predictions from Presto's pixel outputs (illustrated in Figure 6):

1. We encode the pixels in an image individually, yielding $N$ output tokens.
2. We then measure the mean and standard deviation of these $N$ output pixels per dimension and concatenate the result, yielding a $2d_e$ vector (where, $d_e$ is Presto's output token size, or 128).
3. We then pass this mean and standard deviation vector to a downstream classifier.

We evaluate Presto using three image-based datasets:

- **TreeSatAI**: The TreeSatAI dataset consists of detecting the presence of one or more tree species (out of 20 possible species) in forestry images in Germany (Ahlswede et al., 2023). We use the train and test splits provided by Ahlswede et al. (2023), and compare Presto to the deep learning and tree-based baselines provided alongside the dataset. As done for the baselines, we measure the effectiveness of models using only Sentinel-2 (S2) or only Sentinel-1 (S1) data.
- **EuroSAT**: The EuroSAT dataset consists of classifying Sentinel-2 multispectral images in Europe as belonging to one of 10 landcover classes (Helber et al., 2019). We use the train and test splits provided by Neumann et al. (2019). We compare Presto to SatMAE, ConvMAE and ScaleMAE by using the kNN-classifier approach at a variety of input resolutions, as is done by Reed et al. (2022). We also compare finetuned Presto against Seasonal Contrast (Manas et al., 2021) and GASSL (Ayush et al., 2021). Since most models ingest RGB imagery, we evaluate Presto both when it receives all Sentinel-2 bands as input (MS) and when it only receives the RGB bands.

## 4 RESULTS

Overall, Presto excels at a variety of tasks in a range of geographies (spanning 4 continents and 38 countries). Presto is performant both when the whole model is fine-tuned (Table 5) or and when it is used as a feature extractor for simple models (Tables 3, 4 and 3). We highlight that wherever it is possible to do so, we benchmark Presto against the state-of-the-art model for that task – even as a feature extractor, Presto outperforms this state-of-the-art model in many cases (Tables 3 and

| Model | Backbone | Finetuned params (M) | Finetuning accuracy |
|---|---|---|---|
| GASSL | ResNet-18 | 11.69 | 0.895 |
| SeCo | ResNet-18 | 11.69 | 0.931 |
| SatMAE (RGB) | ViT-Large | 303.10 | 0.955 |
| SatMAE (MS) | ViT-Large | 305.96 | 0.990 |
| Presto (RGB) | Presto | 0.40 | 0.826 |
| Presto (MS) | Presto | 0.40 | 0.944 |

(a)

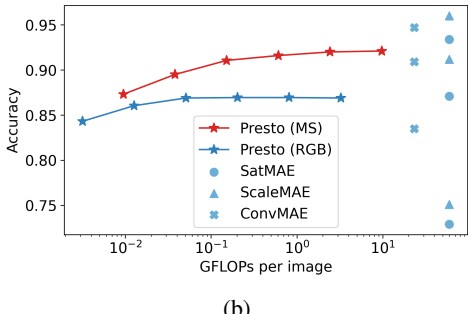

(b)

Figure 3: **Presto is competitive with methods designed for single-timestep satellite images, while being much more computationally efficient**. We show (a) EuroSAT finetuning accuracy for a variety of architectures and pre-training regimes, and (b) accuracy of the embeddings when passed through a KNN@5 classifier at a variety of input resolutions (following (Reed et al., 2022)) as a function of FLOPs required to encode an image (note the log scale on the x-axes). In (b), all image-based models resize images to $224 \times 224$, so the FLOPs required to encode an image do not change with image resolution. Presto achieves competitive results with image-based models while requiring **up to four orders of magnitude less FLOPs to encode an image**.

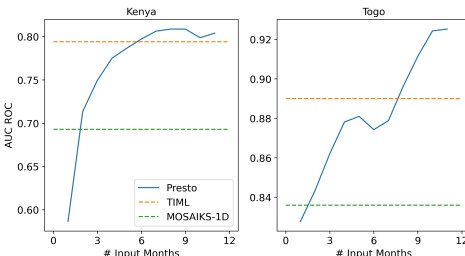

Figure 4: **Presto is robust to incomplete inputs**. We measure the AUC ROC score of Presto with Linear probing (Presto$_R$) on the CropHarvest dataset when no Dynamic World input is passed, and with a subset of input months (the x-axis). Presto$_R$ recovers the performance of MOSAIKS-1D and TIML with 12 months of input given only a subset of input months.

4). In addition, while previous self-supervised learning approaches primarily consider downstream land-cover mapping tasks we demonstrate Presto's performance across a wide range of tasks ranging from tree species classification to algae bloom estimation.

**Applying Presto to downstream tasks is computationally efficient**  While other methods require a cluster of GPUs at fine-tuning time (Cong et al., 2022), we fine-tune Presto on a single GPU or CPU (Presto can be fine-tuned on the Fuel Moisture task on a 2017 MacBook Pro in less than 6 minutes). When Presto is used as a feature extractor, only simple models need to be trained (which require few parameters to be learned, as we show in Table 3). Even when fully fine-tuned, Presto's small size means that relatively few parameters need to be finetuned (Figure 3(a)). This makes Presto accessible to practitioners, especially those lacking significant computational resources.

**Presto is performant even with missing data**  Presto is pre-trained on a variety of diverse remote sensing products (see Section 2.1). However, these data products may not be available for a downstream task; for example, only a single timestep or single sensor may be available (Helber et al., 2019; Ahlswede et al., 2023). We therefore evaluate Presto when it receives:

- **A subset of timesteps**: We evaluate Presto when it receives only a subset of timesteps compared to the 12 timesteps used for pre-training. Specifically, Presto receives 3 input timesteps for the fuel moisture task (Table 5) and only a single input timestep for the EuroSat and TreeSatAI tasks (Tables 4 and 3). In all cases, Presto remains performant. We also evaluate Presto when a subset of input months are passed for the CropHarvest dataset (Figure 4) – Presto rapidly recovers the performance of TIML and MOSAIKS-1D (which used all input months).
- **A subset of input bands**: We evaluate Presto when it only receives a subset of the training input bands. For the EuroSat task (Table 3), Presto receives either the full Sentinel-2 input, or only the RGB bands (which represents only a single token, since only one timestep is available for this task). Similarly, we evaluate Presto when it receives either Sentinel-2 or Sentinel-1 data for the

Table 5: RMSE results on the regression tasks. While the literature baselines are not directly comparable, since they use different input datasets or private test data (or both), Rao et al. (2020) report an RMSE of 25 on the fuel moisture dataset with a physics-assisted neural network and the algae bloom competition winner reported an RMSE of 0.761, indicating our results are within the scope of utility. Best results are **highlighted blue**, with second best results in **bold**. Results are an average of three runs with standard error reported. Models have a high variance in performance across tasks – we therefore calculate the mean difference in RMSE from the linear regression baseline across both tasks. Presto performs most consistently, both when used as a feature-extractor and when fine-tuned.

|  | Fuel Moisture | Algae Blooms | Mean difference |
|---|---|---|---|
| Linear Regression | 28.20 | **0.850** | 0% |
| Random Forest | **23.84 ± 0.42** | $1.249 \pm 0.02$ | 15.7% |
| MOSAIKS-1D$_{RF}$ | $28.75 \pm 0.15$ | $0.972 \pm 0.01$ | 8.15% |
| Presto$_{FT}$ (random init.) | $26.07 \pm 0.52$ | $0.955 \pm 0.05$ | 2.40% |
| Presto$_{FT}$ | **25.28 ± 0.30** | **0.815 ± 0.03** | **−7.24%** |
| Presto$_{RF}$ | $25.98 \pm 0.66$ | $0.884 \pm 0.01$ | **−1.94%** |

TreeSatAI task (Table 4). In both cases, Presto is competitive with methods designed to ingest single-timestep, single-sensor data. Finally, also on the CropHarvest dataset (Table 3), Presto remains performant (compared to a full input) without being passed the Dynamic World input, with a negligible difference in mean F1 score.

**Presto is competitive even in out-of-domain settings against larger models** Because Presto is pre-trained on pixel-timeseries with no spatial context, we consider the image-based tasks (Section 3.2) as out-of-domain. However, image-based datasets are commonly used as remote sensing benchmarks, motivating us to measure Presto's performance in this regime. We find that Presto performs comparably to models specifically designed and trained to ingest image datasets (Table 4 and Figure 3). With a KNN classifier, Presto performs comparably to significantly larger ViT based models on the EuroSAT dataset (Figure 3). Specifically, Presto achieves comparable average accuracy (across the different resolutions) to larger ViT based models when given RGB data, and significantly outperforms these models when given multispectral (MS) data. In addition, Presto requires orders of magnitude less compute to encode the images in both cases and for any resolution. When finetuned, Presto with multi-spectral data performs competitively with remote sensing models designed for images, despite having significantly less parameters to finetune.

## 5 DISCUSSION & CONCLUSION

**Limitations** Presto is designed to ingest 10m/px resolution imagery, and is pre-trained on products at this scale. This decision is motivated by the free, global availability over time of products at this scale (such as Sentinel-1 and Sentinel-2). This means that Presto does not process very high resolution imagery natively. In addition, Presto is a pixel-timeseries model. While we demonstrate Presto's ability on single-timestep image datasets, if it is desirable to process entire images to make a prediction then image-based models which do this natively may be better suited. We highlight the plateauing of Presto's performance on the EuroSat dataset as the input resolution increases (Table 3), due to classes in the dataset where the relevant pixels for the class are a minority of the pixels in the image; in such scene classification challenges, image-based models (which can distinguish the shape of the relevant pixels) may be better suited. We discuss this further in Section A.7.

**Conclusion** We present Presto: a lightweight, pre-trained timeseries transformer for remote sensing. By leveraging structure unique to remote sensing data (specifically, (i) an important temporal dimension, (ii) associated metadata and (iii) a diversity of sensors), we are able to train an extremely lightweight model which achieves state-of-the-art results in a wide variety of globally distributed evaluation tasks. Computational efficiency is of paramount importance in remote sensing settings (often dictating which models ultimately get selected for deployment). We demonstrate that strong performance can be achieved while meeting this constraint, and that self-supervised learning can provide significant benefits even for small models.

ETHICS STATEMENT

Presto is designed to be accessible to a wide range of practitioners; we achieve this by only training Presto on publicly available data and by keeping the model size small enough so it can be leveraged in compute-constrained environments. In addition to increasing Presto's accessibility, its small compute footprint also lowers its carbon footprint (Strubell et al., 2019).

As described by Tuia et al. (2023), a natural concern when applying machine learning algorithms to remote sensing data is its use to collect information about individuals who are unaware that data is being collected, and therefore cannot consent to this practice. We therefore encourage deployment of Presto in collaboration with local communities and stakeholders (Krafft; Kshirsagar et al., 2021; Nakalembe & Kerner, 2023).

REPRODUCIBILITY

All code and data used to train and evaluate Presto will be made available upon publication, and the code is currently available anonymously at `https://anonymous.4open.science/r/presto-C4AB`. In addition, we discuss specific implementation details in Appendices A.1 and A.4. We have strived to make the Presto codebase accessible to other practitioners; to this end, we include a demo Jupyter notebook demonstrating how Presto can be applied to a new downstream task, which is available anonymously at `https://anonymous.4open.science/r/presto-C4AB/downstream_task_demo.ipynb`.

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

## A   APPENDIX

### A.1   PRE-TRAINING DETAILS

We outline training hyperparameters below:

- **Training length**: We train the model for 20 epochs, with a batch size of 4096 (resulting in 5950 batches per epoch). On a single NVIDIA V100 GPU, this takes $43\frac{1}{4}$ hours.
- **Optimizer and learning rate**: We train the model with an AdamW optimizer. We use a cosine annealing schedule for our learning rate, with a maximum learning rate of 0.001 at the 2nd epoch. We apply a weight decay of 0.05, and $\beta$s of (0.9, 0.95).
- **Masking**: We use a masking ratio of 0.75, randomly selecting (for each instance) a masking strategy from the ones described in Section 2.3. If the masking strategy cannot mask the right number of tokens, we randomly mask additional tokens to achieve the correct masking ratio.

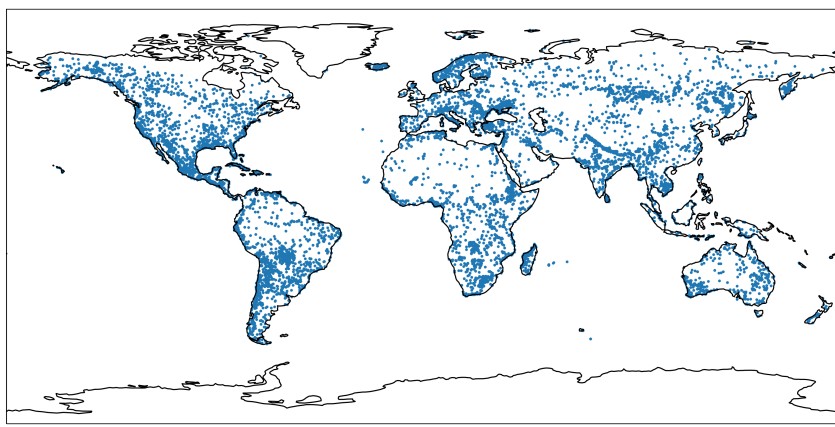

Figure 5: The distribution of the pre-training dataset described in Section 2.1.

### A.1.1 PRETRAINING DATA

Remote sensing models can be deployed in a wide range of geographies, with few labelled datapoints available at fine-tuning time (Kerner et al., 2020; Böhm et al., 2022). We therefore aim to collect a globally representative pre-training dataset. We achieve this by following the sampling strategy used by Dynamic World (Brown et al., 2022). We divide the Earth into three regions: the Western Hemisphere and two regions in the Eastern Hemisphere. These regions are further divided into ecoregions, and stratified samples are gathered from each region using land cover classes as sampling strata. Figure 5 shows the resulting geographical distribution. Each sample represents a $510 \times 510$ pixel tile with a spatial resolution of 10 meter per pixel. To obtain pixel-timeseries we grid-sample 2,500 pixels from each sample, yielding a total of 21,535,000 pixel samples (each with 24 one-month timesteps).

### A.1.2 CHANNEL GROUPS

As described in Section 2.2, we transform the pixel timeseries $x$ into a number of tokens, where each token is a linear transformation of a subset of the input channels. We group together channels which (i) come from the same sensor or product, (ii) have equivalent native spatial resolutions and (iii) represent similar parts of the electromagnetic spectrum (for Sentinel-2 channel groups). We group the input data into the following channel groups:

- **Sentinel-1**: The VV and VH bands from the Sentinel-1 sensor
- **Sentinel-2 RGB**: The B2, B3 and B4 bands from the Sentinel-2 sensor
- **Sentinel-2 Red Edge**: The B5, B6 and B7 bands from the Sentinel-2 sensor
- **Sentinel-2 Near Infra Red (10m)**: The B8 band from the Sentinel-2 sensor
- **Sentinel-2 Near Infra Red (20m)**: The B8A band from the Sentinel-2 sensor
- **Sentinel-2 Short Wave Infra Red**: The B11 and B12 bands from the Sentinel-2 sensor
- **NDVI**: The normalized difference vegetation index, calculated from the Sentinel-2 B4 and B8 bands.
- **ERA5 Climatology**: Precipitation and temperature at 2m from the ERA5 Climate Reanalysis product
- **Topography**: The elevation and slope of a pixel, calculated by the SRTM's DEM
- **Location**: The cartesian coordinates of a pixel, computed from the pixel's latitude and longitude

### A.2 FLOP CALCULATIONS

We use the `thop` library (https://github.com/Lyken17/pytorch-OpCounter) to calculate the FLOPs required to encode a EuroSAT image (as plotted in Table 3(b)). For the SatMAE, ScaleMAE and ConvMAE models, all images were resized to $224 \times 224$, so the FLOPs required to encode an image is independent of resolution. For Presto, we computed the FLOPs required to encode a single pixel and multiplied this by the number of pixels in an image at each resolution (e.g.

Table 6: FLOPs required to encode a single EuroSat image (or pixel, for Presto), as measured by the `thop` library. When plotting results in Table 3, we multiply the FLOPs for Presto by the number of pixels encoded for an image. At its highest resolution, EuroSAT images are $64 \times 64$, so Presto FLOPs for a full resolution image can be obtained by multiplying the per-pixel FLOPs by 4,096. We include this value in brackets for completeness.

| Model | Backbone | MegaFlops |
|---|---|---|
| SatMAE (RGB) (Cong et al., 2022) | ViT-Large | 59,685.69 |
| SatMAE (MS) (Cong et al., 2022) | ViT-Large | 535,515.25 |
| ScaleMAE (Reed et al., 2022) | ViT-Large | 59,685.69 |
| ConvMAE (Gao et al., 2022) | ConvMAE-Large | 23,315.58 |
| SeCo (Manas et al., 2021) | ResNet-18 | 149.37 |
| GASSL (Ayush et al., 2021) | ResNet-18 | 149.37 |
| Presto RGB pixel (image) | Presto | 0.79 (3,235.84) |
| Presto MS pixel (image) | Presto | 2.37 (9,707.52) |

the "64" resolution has $64 \times 64$ pixels, so we multiply the FLOPs required to encode a single pixel by $64 \times 64 = 4096$). The FLOPs calculated by the `thop` library are recorded in Table 6.

## A.3 BASELINES

In addition to the task-specific baselines described above, we benchmark Presto against:

- **Random Forests**: Random forests are powerful baselines in remote sensing as they they remain competitive with state-of-the-art methods (Pelletier et al., 2019; Kerner et al., 2020). Tree-based methods, especially random forests, are commonly deployed in large-scale machine learning for remote sensing applications (Hansen et al., 2013; Van Tricht, 2021; Di Tommaso et al., 2022).
- **MOSAIKS-1D**: We adapt MOSAIKS (Rolf et al., 2021) for timeseries data. MOSAIKS-1D uses patches from the pre-training dataset and convolves over the temporal dimension instead of the spatial dimension. We benchmark MOSAIKS-1D on all timeseries evaluation tasks. Because this does not work for categorical inputs, we exclude Dynamic World. As with Presto, we use the output features with random forests (MOSAIKS-1D$_{RF}$) and with regressions (MOSAIKS-1D$_R$).
- **Fully Supervised Presto**: To disentangle the effects of the model architecture from the pre-training regimen, we fine-tune a Presto architecture starting from randomly initialized weights.

## A.4 DOWNSTREAM RESULTS

We include complete results for the evaluation tasks. These include error bars, as well as additional results reported for the CropHarvest (Table 9), EuroSAT (Tables 10 and 11) and TreeSatAI datasets (Table 12).

We run all downstream classifiers with 3 seeds $(0, 42, 84)$, with the exception of the KNN classifiers and the linear regression (which are deterministic). In the tables in the main paper (Tables 3, 4 and 5) we report the average of these runs; the standard error is reported in Tables 9,12 and 5.

- **Presto as a feature extractor**: When used as a feature extractor, a random forest, regression of K-nearest-neighbours classifier is trained on Presto's output embeddings. In this case, we use scikit-learn models with the default hyperparameters. The CropHarvest tasks, the class labels are extremely balanced; we therefore set `class_weight` equal to `balanced` for those tasks, for both Presto and MOSAIKS-1D.
- **Fine-tuning Presto**: When fine-tuning Presto, we use the same hyperparameters across all tasks: an AdamW optimizer with a learning rate of `3e-4` and a batch size of 64. We use a geographically seperated validation set with early stopping, with a patience of 10.

As discussed in Section 3.2, we obtain per-image predictions using Presto by computing a mean and standard deviation of Presto's output pixels, and passing a concatenation of these two vectors to a downstream classifier. This is illustrated in Figure 6.

Table 7: Accuracy results for pre-trained and from-scratch Presto when finetuned on EuroSat, at varying resolutions. We hypothesize that the drop in performance for the full resolution (64) RGB input is due to the model construction; the model outputs for all pixels in the image (4,096 pixels for the full resolution) are aggregated and passed to a linear layer for classification, yielding an extremely noisy gradient signal.

| Resolution | | 2 | 4 | 8 | 16 | 32 | 64 |
|---|---|---|---|---|---|---|---|
| random init. | RGB | 0.644 | 0.745 | 0.682 | 0.732 | 0.713 | 0.711 |
| pretrained | | 0.804 | 0.801 | 0.838 | 0.863 | 0.866 | 0.826 |
| random init. | MS | 0.759 | 0.869 | 0.857 | 0.883 | 0.927 | 0.914 |
| pretrained | | 0.879 | 0.906 | 0.934 | 0.937 | 0.943 | 0.944 |

Table 8: **Effect of model size on validation performance**. To understand the effect of model size on performance, we pre-train two larger variants of Presto. As in Table 1, we measure $Presto_R$'s performance on the CropHarvest validation task. In this table, the number of parameters includes both the number of encoder and decoder parameters. The FLOPS are computed for a "full" input (12 timesteps, with no missing channels), when passed through both the encoder and decoder.

| Model | Encoder | | Decoder | | # params (M) | FLOPS (M) | F1 score |
|---|---|---|---|---|---|---|---|
| | Depth | Width | Depth | Width | | | |
| Presto | 2 | 128 | 2 | 128 | 0.81 | 88.94 | 0.665 |
| Presto Wide | 2 | 256 | 2 | 128 | 2.02 | 220.81 | 0.687 |
| Presto Deep | 4 | 128 | 2 | 128 | 1.21 | 132.42 | 0.669 |

## A.5 DISENTANGLING THE EFFECT OF PRE-TRAINING

To understand the effect of pre-training Presto, we finetune Presto and train it from scratch on EuroSat (Table 7), the regression tasks (Table 5 in the main paper) and TreeSatAI (Table 12). We omit the CropHarvest dataset because it was expressly designed as a few-shot-learning dataset. Its small size makes the construction of validation sets with which to control the finetuning (e.g. with early stopping) challenging.

Overall, we find a consistent and significant improvement from the use of pre-trained Presto compared to a randomly initialized version of the model. For the EuroSat task, pre-training consistently delivers an incresse in accuracy score $> 0.1$ (representing increases in accuracy of up to 25%). This effect is consistent with what we observe on the TreeSatAI dataset (where pretraining consistently increases mAP by around 10 points) and on the regression tasks (where pretraining reduces RMSE by to 15% on the algae blooms task).

## A.6 MODEL SIZE EFFECTS

To measure how different model sizes affect Presto's performance, in addition to the base model we pre-train two larger variants of Presto - a deeper variant with 4 layers in the encoder instead of 2, and one wider variant with an encoder-size of 256 instead of 128. We show these results in Table 8.

Performance improves as model size increases, suggesting that practitioners who are less sensitive to the performance / compute trade off could obtain better results by training a larger Presto model.

## A.7 PRESTO'S FAILURE MODES

Presto processes pixel-timeseries independently, without spatial context from other pixels or locations. This means that when we make image-based predictions (such as for scene classification), Presto's independent pixel representations must be aggregated into a single prediction. We opt for a simple concatenation of the element-wise mean and standard deviation of the representations, from

Table 9: Additional results for the CropHarvest task. In addition to the F1 scores reported in the main paper, we report AUC ROC scores, with standard error bars computed with three runs.

| | Model | Kenya | Brazil | Togo | Mean |
|---|---|---|---|---|---|
| F1 | Random Forest | $0.559 \pm 0.003$ | $0.000 \pm 0.000$ | $0.756 \pm 0.002$ | 0.441 |
| | MOSAIKS-1D$_R$ | $0.790 \pm 0.027$ | $0.746 \pm 0.084$ | $0.679 \pm 0.024$ | 0.738 |
| | TIML | $0.838 \pm 0.000$ | $0.835 \pm 0.012$ | $0.732 \pm 0.002$ | 0.802 |
| | Presto$_R$ | $0.816 \pm 0.000$ | $\mathbf{0.891 \pm 0.000}$ | $\mathbf{0.798 \pm 0.000}$ | 0.835 |
| | no DW | $\mathbf{0.861 \pm 0.000}$ | $0.888 \pm 0.000$ | $0.760 \pm 0.000$ | **0.836** |
| AUC ROC | Random Forest | $0.578 \pm 0.006$ | $0.941 \pm 0.004$ | $0.892 \pm 0.001$ | 0.803 |
| | MOSAIKS-1D$_R$ | $0.693 \pm 0.036$ | $0.890 \pm 0.038$ | $0.836 \pm 0.005$ | 0.806 |
| | TIML | $0.794 \pm 0.003$ | $0.988 \pm 0.001$ | $0.890 \pm 0.000$ | 0.890 |
| | Presto$_R$ | $0.834 \pm 0.000$ | $\mathbf{0.997 \pm 0.000}$ | $\mathbf{0.921 \pm 0.000}$ | 0.917 |
| | no DW | $\mathbf{0.863 \pm 0.000}$ | $0.989 \pm 0.000$ | $0.912 \pm 0.000$ | **0.921** |

Table 10: Additional results for the EuroSat task - results for the ScaleMAE, SatMAE and ConvMAE models are from Reed et al. (2022). We report KNN classifier results for different values of $k$, and at varying input resolutions.

| Resolution | 16 | | | 32 | | | 64 | | |
|---|---|---|---|---|---|---|---|---|---|
| $k$ | 5 | 20 | 100 | 5 | 20 | 100 | 5 | 20 | 100 |
| SatMAE | 0.729 | 0.727 | 0.695 | 0.871 | 0.876 | 0.854 | 0.934 | 0.931 | 0.913 |
| ScaleMAE | 0.751 | 0.744 | 0.699 | 0.912 | 0.901 | 0.869 | **0.960** | 0.956 | 0.935 |
| ConvMAE | 0.835 | 0.826 | 0.788 | 0.909 | 0.898 | 0.863 | 0.947 | 0.940 | 0.914 |
| Presto (RGB) | 0.869 | 0.828 | 0.713 | 0.869 | 0.829 | 0.712 | 0.869 | 0.829 | 0.713 |
| Presto (MS) | **0.916** | 0.892 | 0.844 | **0.920** | 0.892 | 0.846 | 0.921 | 0.893 | 0.846 |

Table 11: Additional results for the EuroSat task for Presto when run with reduced resolutions (compared to those used by Reed et al. (2022) and reported in Table 10). We report KNN classifier results for different values of $k$, and at varying input resolutions.

| Resolution | 2 | | | 4 | | | 8 | | |
|---|---|---|---|---|---|---|---|---|---|
| $k$ | 5 | 20 | 100 | 5 | 20 | 100 | 5 | 20 | 100 |
| Presto (RGB) | 0.843 | 0.811 | 0.699 | 0.860 | 0.820 | 0.706 | 0.869 | 0.826 | 0.710 |
| Presto (MS) | 0.873 | 0.852 | 0.799 | 0.895 | 0.874 | 0.824 | 0.911 | 0.886 | 0.838 |

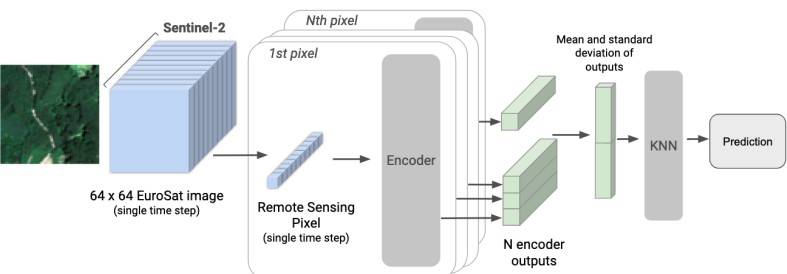

Figure 6: We obtain per-image predictions using Presto by computing a mean and standard deviation of Presto's per-pixel outputs, and passing this concatenated vector to a downstream classifier. We illustrate what this process looks like for the EuroSat task.

Table 12: Additional results for the TreeSatAI (as in Ahlswede et al. (2023), we report precision and recall in addition to $F_1$ score and mAP). In addition, we report the results of finetuning Presto ($\text{Presto}_{FT}$) from the pre-trained weights and from a random initialization.

| Model | Data | Aggregation | $F_1$ | mAP | Precision | Recall |
|---|---|---|---|---|---|---|
| MLP | | | 10.09 | 29.42 | 33.29 | 7.13 |
| LightGBM | | | 11.86 | 32.79 | 37.96 | 8.06 |
| $\text{Presto}_{FT}$ (random init.) | | Weighted | $29.61 \pm 0.13$ | $26.83 \pm 0.39$ | $20.90 \pm 0.21$ | $65.26 \pm 0.19$ |
| $\text{Presto}_{FT}$ | | | $31.95 \pm 0.09$ | $35.43 \pm 0.12$ | $19.90 \pm 0.07$ | $92.08 \pm 0.29$ |
| $\text{Presto}_{RF}$ | S1 | | $19.79 \pm 0.00$ | $35.76 \pm 0.00$ | $51.90 \pm 0.02$ | $14.16 \pm 0.00$ |
| MLP | | | 12.82 | 33.09 | 63.01 | 7.13 |
| LightGBM | | | 14.07 | 35.11 | 55.49 | 8.06 |
| $\text{Presto}_{FT}$ (random init.) | | Micro | $32.08 \pm 0.50$ | $28.79 \pm 0.48$ | $21.28 \pm 0.44$ | $65.26 \pm 0.19$ |
| $\text{Presto}_{FT}$ | | | $28.10 \pm 0.22$ | $38.94 \pm 0.10$ | $16.59 \pm 0.17$ | $92.08 \pm 0.29$ |
| $\text{Presto}_{RF}$ | | | $22.92 \pm 0.00$ | $38.69 \pm 0.00$ | $60.17 \pm 0.00$ | $14.16 \pm 0.00$ |
| MLP | | | 51.97 | 64.19 | 74.59 | 42.23 |
| LightGBM | | | 48.17 | 61.99 | 74.27 | 40.04 |
| $\text{Presto}_{FT}$ (random init.) | | Weighted | $31.29 \pm 0.17$ | $39.82 \pm 1.34$ | $19.05 \pm 0.16$ | $87.74 \pm 0.63$ |
| $\text{Presto}_{FT}$ | | | $37.49 \pm 0.37$ | $53.43 \pm 0.77$ | $24.40 \pm 0.29$ | $90.77 \pm 0.29$ |
| $\text{Presto}_{RF}$ | S2 | | $46.26 \pm 0.00$ | $60.88 \pm 0.00$ | $75.42 \pm 0.00$ | $37.08 \pm 0.00$ |
| MLP | | | 54.49 | 65.83 | 77.18 | 42.23 |
| LightGBM | | | 52.52 | 61.66 | 76.27 | 40.04 |
| $\text{Presto}_{FT}$ (random init.) | | Micro | $35.68 \pm 0.16$ | $39.14 \pm 1.01$ | $23.35 \pm 0.19$ | $87.74 \pm 0.63$ |
| $\text{Presto}_{FT}$ | | | $33.19 \pm 0.38$ | $53.06 \pm 1.21$ | $20.30 \pm 0.27$ | $90.77 \pm 0.29$ |
| $\text{Presto}_{RF}$ | | | $50.41 \pm 0.00$ | $63.24 \pm 0.00$ | $78.70 \pm 0.00$ | $37.08 \pm 0.00$ |

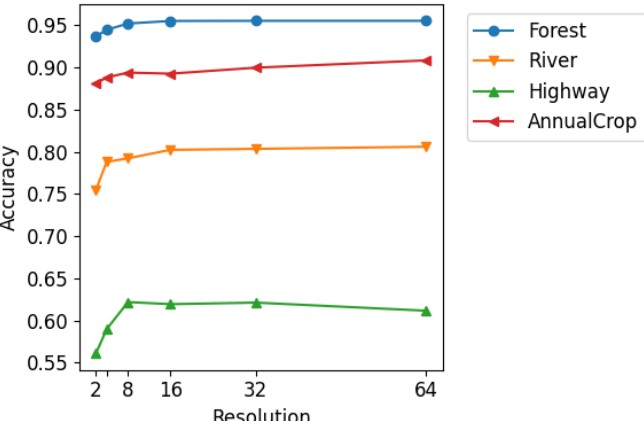

Figure 7: Accuracy of KNN@5 classifier with Presto RGB representations on the EuroSat dataset vs. the input resolution, for different categories. Some categories have been left out for clarity.

which a classifier makes a prediction. Information gets lost in such a simple aggregation, which impacts Presto's performance on such tasks.

For example, Presto's performance on the EuroSat dataset reaches a plateau when increasing the input resolution. As Figure 7 shows, this is mainly caused by a failure to accurately predict specific classes (for example, the *Highway* and *River* classes). Figure 8 shows example images for these classes, as well as for the *Forest* and *AnnualCrop* classes, on which Presto achieves higher accuracies. While in the *Forest* and *AnnualCrop* images, most pixels of the image actually represent the labelled class, in the *Highway* and *River* images only a relatively small part of the image actually contains the label (a highway or river). We hypothesize that since many pixels in the *Highway* and *River* images do not actually represent that class, the crude token-aggregation method we use to represent images is insufficiently discriminative to accurately classify these images.

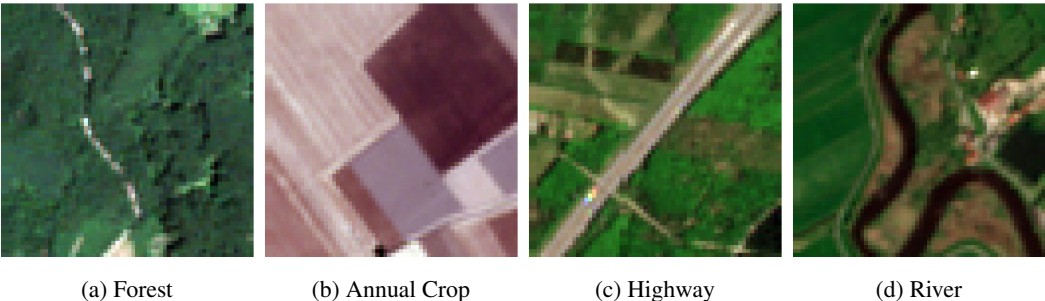

(a) Forest          (b) Annual Crop          (c) Highway          (d) River

Figure 8: the RGB bands of example images from EuroSat classes.

Other pre-trained remote sensing models use much more powerful mechanisms for aggregating spatial information. For example, ViT models convolve over patches and then apply an attention mechanism between spatial patches. If image-based predictions are needed and these predictions are highly dependent on the occurrence of objects in subregions of the image, models which natively process this important spatial information may be better suited.

We hypothesize that lossy aggregation of spatial information also penalizes Presto on the TreeSatAI dataset; when passed S2 data, it performs worse than the MLP and LightGBM models, which receive as input concatenated pixel inputs (Table 12).

We plan on exploring techniques to mitigate this difficulty with Presto in future work.

