# OpenReview forum: "Lightweight, Pre-trained Transformers for Remote Sensing Timeseries"
_ICLR.cc/2024/Conference — Submitted to ICLR 2024_

### Official Review · Reviewer_X32p · 2023-10-28

**Soundness:** 3 good
**Presentation:** 3 good
**Contribution:** 3 good
**Rating:** 5
**Confidence:** 4

**Summary:**

The authors introduce the Pretrained Remote Sensing Transformer (Presto) to address challenges in parsing remote sensing data with machine learning, emphasizing the unique temporal and sensor diversity attributes of this data. Presto diverges from traditional self-supervised learning approaches, which often treat remote sensing similarly to natural imagery. Presto is designed to be adaptable across various Earth observation sensors, remains robust despite missing data, and showcases computational efficiency, particularly when compared to models like ViT or ResNet.

**Strengths:**

##### S1: Tailored for Remote Sensing Data:
Unlike other models that treat remote sensing data akin to natural imagery, Presto is explicitly designed for remote sensing's unique characteristics, such as the importance of the temporal dimension and the variety of sensors involved.

##### S2: Computational Efficiency:
Presto, despite its competitive performance, has a significantly smaller model size compared to ViT or ResNet models. This makes it suitable for real-world deployment, especially when processing large volumes of data to make geospatial predictions.

##### S3: The paper is well-written and organized

The paper presents its results in a clear and organized, making it accessible and informative for readers.

**Weaknesses:**

##### W1: Understanding the proposed method:

While the proposed methods are effective and efficient, I'd expect more discussions or derivation to illustrate the reason to give insights to understand the framework better.


##### W2: Literature reviews and comparison of related work:

Presto is designed to ingest 10m/px resolution imagery and designed to fuse temporal information. I'd suggest authors have further literature review or experiences. For example, for pre-training and downstream tasks, there is a dataset like Extended AgVision[1] with 10m/px resolution and labeled/unlabeled imageries on large scales. For comparison, authors could compare with the backbones of pretrained from seasonal contrast [2] that authors have discussed in the paper. I'd suggested authors give a more comprehensive review and comparison in the related work and experiments section.

[1] Extended Agriculture-Vision: An Extension of a Large Aerial Image Dataset for Agricultural Pattern Analysis

[2] Seasonal Contrast: Unsupervised Pre-Training from Uncurated Remote Sensing Data

**Questions:**

The main questions are listed in Weaknesses. I'd raise my score if they were appropriately addressed.

---

> ### Author Response · Authors · 2023-11-17
> **Response to Reviewer X32p**
>
> Thank you for your review.
>
> - **Understanding the proposed method**
>
>   Thank you for your suggestion. We would be happy to clarify our framework and our motivations - are there any specific questions you have about the framework which we could answer either in this response or in the paper itself?
>
>   For clarity, we include some of the motivations and descriptions we currently include in the paper:
>
>   In the introduction (Section 1), we motivate the method by describing the following attributes of remote sensing, and desired characteristics:
>
>   - **The ingestion of highly multi-modal data**: We aim to leverage the different modalities of data being captured by satellites around the Earth. One of the ways we achieve this is by introducing learned encodings to the transformer method which specifically communicate to the model which sensors a token is constructed from (we describe this in Section 2.2). In addition, our model ingests both raw (e.g. optical) and derived (e.g. landcover-maps) products (we describe this in section 2.1). We hypothesize that this allows the model to map raw data to semantic meaning during the pre-training phase.
>
>   - **A strong emphasis on the temporal dimension**: Unlike comparable MAE methods (such as ScaleMAE or SatMAE) designed for remote sensing data which treat the data as images (and therefore emphasise the spatial dimension), our model ingests pixel timeseries. We do this because for many remote sensing tasks, measuring change over time is critical; modelling remote sensing data as timeseries is therefore a common practice among remote sensing practitioners.
>
>   - **An emphasis on computational efficiency**: As described in Section 1 (and elaborated in our response to reviewer DF48, ([link](https://openreview.net/forum?id=Iip7rt9UL3&noteId=oZLo98iQY2)), computational efficiency is critical to manage costs when deploying remote sensing models. We therefore keep Presto very lightweight (something which is in part possible because timeseries inputs have a much lower dimensionality than images).
>
>   Figure 1 illustrates the specific pre-training pipeline, with an emphasis on demonstrating the use of diverse inputs (which are both dynamic and static in time), and the masking strategies we implement to support a range of downstream tasks.
>
>   We also include more details about the Presto architecture in Table 8.
>
> - **Literature reviews and comparison of related work.**
>
>   Thank you for suggesting the AgVision dataset. It is at a 10cm/pixel (centimetres, and not metres) resolution, which Presto is not trained to ingest (this is a factor 100 difference with our pre-training data). We discuss this limitation in Section 5 of our paper.
>
>   In addition, thank you for suggesting the Seasonal Contrast Paper - we have added a comparison to it in Figure 3a. In addition, we also compare to Geography-Aware Self-Supervised Learning [1] in the same table.
>
>   In general, we have updated Section 1 to include more discussion of related works, including Transformer-based modelling for remote sensing time series. Thank you for this suggestion.
>
> [1] Ayush, Kumar, et al. "Geography-aware self-supervised learning." Proceedings of the IEEE/CVF International Conference on Computer Vision. 2021.

---

### Official Review · Reviewer_T6WZ · 2023-10-29

**Soundness:** 2 fair
**Presentation:** 3 good
**Contribution:** 2 fair
**Rating:** 5
**Confidence:** 4

**Summary:**

The paper introduces a lightweight, pre-trained transformer model called Presto for analyzing remote sensing data. The authors highlight the challenge of acquiring labeled data for training remote sensing models and the need for self-supervised learning approaches. Presto is designed to leverage the temporal dimension and data from multiple sensors in remote sensing data. The model is pre-trained on a diverse range of Earth observation products and achieves competitive performance with state-of-the-art models while requiring significantly less computing. The main contributions of the paper are the introduction of Presto, its competitive performance across various tasks and geographies, and its efficiency in terms of model size and computational requirements.

**Strengths:**

1. Comprehensive Pretraining Data: The paper mentions that Presto is pre-trained on a diverse range of directly sensed and derived Earth observation products, which can significantly improve model performance. This comprehensive pretraining data helps the model capture a wide range of features and patterns in remote sensing data.

2. Competitive Performance: The paper highlights that Presto achieves state-of-the-art results in a wide variety of globally distributed evaluation tasks. It outperforms benchmark models in many cases, even when used as a feature extractor. This indicates that Presto is effective in capturing relevant information from remote sensing data and producing accurate predictions.

3. Lightweight and Efficient: Presto is described as a lightweight model that meets the computational efficiency requirements of remote sensing settings. It has a small compute footprint, making it accessible and deployable in compute-constrained environments. This efficiency is achieved without compromising performance, as Presto performs consistently well across tasks.

4. Self-Supervised Learning Approach: The paper emphasizes the use of self-supervised learning, which allows for training the model on publicly available data. By leveraging self-supervised learning, the model can learn meaningful representations from the data without relying on extensive labeled datasets.

5. Wide Range of Geographies: Presto is shown to perform well across a range of geographies, spanning four continents and 38 countries. This indicates that the model is robust and generalizable, making it suitable for analyzing remote sensing data from various regions around the world.

**Weaknesses:**

1.	Limited Spatial Context: Presto is designed to process pixel-time series data and does not process very high-resolution imagery natively. This limitation may impact its performance on tasks where spatial information is crucial, such as scene classification challenges. Image-based models that can distinguish the shape of relevant pixels may be better suited for such tasks.
2.	Lossy Aggregation of Spatial Information: The paper mentions that Presto uses a crude token-aggregation method to represent images, which may result in a loss of spatial information. This lossy aggregation method can penalize Presto's performance on tasks that heavily rely on the occurrence of objects in subregions of the image. Future work is planned to address this difficulty with Presto.
3.	Plateauing Performance with Increasing Input Resolution: The performance of Presto on the EuroSat dataset reaches a plateau as the input resolution increases. This plateauing is mainly caused by a failure to accurately predict specific classes, such as the Highway and River classes. This suggests that Presto may struggle with accurately classifying images where the relevant pixels for a class are a minority of the pixels in the image.
4.	Lack of Discussion on Specific Limitations: While the paper briefly mentions limitations related to spatial context and lossy aggregation of spatial information, it does not provide an in-depth discussion of these limitations. Further exploration and mitigation techniques for these limitations are mentioned as future work. A more detailed discussion of these limitations would provide a clearer understanding of the model's constraints.

**Questions:**

a. How does Presto handle missing or unreliable labels in remote sensing datasets? Are there any specific techniques or strategies employed to mitigate the impact of limited labeled data?
b. Can Presto effectively handle tasks that require a high level of spatial context, such as scene classification challenges? If not, are there any plans to address this limitation in future work?
c. How does Presto perform when the input resolution increases? Are there any specific challenges or limitations observed in accurately predicting specific classes at higher resolutions?
d. Are there any privacy concerns associated with the use of Presto for collecting information about individuals who are unaware that data is being collected? How can these concerns be addressed when deploying Presto in collaboration with local communities and stakeholders?

---

> ### Author Response · Authors · 2023-11-21
> **Response to Reviewer T6WZ**
>
> Thank you for your review. We address your points below.
>
> 1. **Presto’s performance on scene classification tasks**:
>
>     We agree with your points in Weaknesses 1, 2, and 3 (which are concerned with Presto’s processing of pixel-timeseries as opposed to spatial information). While we highlight these weaknesses in our paper, Presto is still performant on scene classification tasks - in particular, we show the performance vs. parameter / FLOPs tradeoff in Figure 3 of the updated paper, where Presto achieves competitive results with state of the art models for the EuroSat task while requiring orders of magnitude less parameters & FLOPs.
>
>      i. **Limited spatial context**: We agree that for scene classification tasks where this spatial context is important, Presto may perform sub-optimally. We discuss this in more detail in Section A.7. However, for many remote sensing tasks spatial context is much less important than temporal context, and it is therefore common for remote sensing practitioners to train pixel-timeseries models [1,2,3,4,5] (we discuss this in Section 1 of our paper). Presto addresses the use case of these practitioners explicitly, while remaining performant on scene classification tasks.
>
>     ii. **Lossy aggregation of spatial information**: We agree that our token aggregation for spatial information is lossy (and bring this point up in Section A.7). As you note in your review, we plan on addressing this limitation in future work.
>
>     iii. **Plateauing performance with Increasing input resolution**: We agree that Presto may struggle with accurately classifying images where the relevant pixels for a class are a minority of the pixels in the image. We explore this in more detail in Section A.7, including investigating performance for specific classes in Figure 7.
>
> 2. **Lack of discussion on specific limitations**:
>
>     As we note in our point above, we have dedicated significant parts of the paper to discussing potential limitations (including over a page discussing Presto’s performance on tasks where spatial context is important in Section A.7, “Presto’s failure modes”). We have also included a limitations section in Section 5 in the main body of the paper, where we discuss constraints such as Presto’s inability to process very high resolution imagery.
>
>     We would be happy to expand these discussions; is there anything you would specifically like us to explore in more detail?
>
> 3. **Q1: How does Presto handle missing or unreliable labels in remote sensing datasets?**
>
>     We pre-train Presto to learn representations of remote sensing data. We show in Table 3 that models with very few parameters can be learned on top of these representations, allowing Presto to be used even when very few labels are available.
>
>     For example, Presto excels at the CropHarvest task (in which as few as 203 labels are available, as shown in Table 2).
>
> 4. **Q2: Can Presto effectively handle tasks that require a high level of spatial context, such as scene classification challenges? If not, are there any plans to address this limitation in future work?**
>
>     Presto is competitive with SoTA models in the EuroSat scene classification task, while requiring orders of magnitude fewer parameters and FLOPs (Figure 3).
>
>     However, as we discuss in Section A.7, improving Presto’s capability even more in this regime is an area of future work.
>
> 5. **Q3. How does Presto perform when the input resolution increases? Are there any specific challenges or limitations observed in accurately predicting specific classes at higher resolutions?**
>
>     In Section A.7 and Figure 7, we discuss Presto’s failure modes. Specifically, we highlight that in scene classification challenges where the predictions are dependent on the occurrence of objects in subregions of the image, Presto’s performance does not improve as input resolution increases.
>
>     Example classes in the EuroSat class (which we plot in Figures 7 and 8) include the river and highway classes.
>
> 6. **Q4: Are there any privacy concerns associated with the use of Presto for collecting information about individuals who are unaware that data is being collected? How can these concerns be addressed when deploying Presto in collaboration with local communities and stakeholders?**
>
>     In our Ethics statement, we note the privacy concerns raised by [6] for all remote sensing models, which can collect information about individuals who are unaware data is being collected.
>
>     We therefore recommend that Presto (and remote sensing models in general) be deployed in collaboration with local communities and stakeholders, as is discussed in more detail by [7], [8] and [9].
>
> Please let us know if these responses have helped address your questions and concerns, or if not what uncertainties remain.

---

> ### Author Response · Authors · 2023-11-21
> **References for Response to Reviewer T6WZ**
>
> [1] Marc Rußwurm, Nicolas Courty, Remi Emonet, Sebastien Lefevre, Devis Tuia, and Romain Tavenard. End-to-end learned early classification of time series for in-season crop type mapping. ISPRS Journal of Photogrammetry and Remote Sensing, 2023.
>
> [2] Vivien Sainte Fare Garnot, Loic Landrieu, Sebastien Giordano, and Nesrine Chehata. Satellite image time series classification with pixel-set encoders and temporal self-attention. CVPR, 2020
>
> [3] Charlotte Pelletier, Geoffrey I Webb, and Franc¸ois Petitjean. Temporal convolutional neural network for the classification of satellite image time series. Remote Sensing, 2019.
>
> [4] Sherrie Wang, Stefania Di Tommaso, Jillian M Deines, and David B Lobell. Mapping twenty yearsof corn and soybean across the us midwest using the landsat archive. Scientific Data, 2020.
>
> [5] Tomislav Hengl, Jorge Mendes de Jesus, Gerard BM Heuvelink, Maria Ruiperez Gonzalez, Milan Kilibarda, Aleksandar Blagoti´c, Wei Shangguan, Marvin N Wright, Xiaoyuan Geng, Bernhard Bauer-Marschallinger, et al. Soilgrids250m: Global gridded soil information based on machine learning. PLoS one, 2017.
>
> [6] Devis Tuia, Konrad Schindler, Beg¨um Demir, Gustau Camps-Valls, Xiao Xiang Zhu, Mrinalini Kochupillai, Saˇso Dˇzeroski, Jan N van Rijn, Holger H Hoos, Fabio Del Frate, et al.

---

### Official Review · Reviewer_DF48 · 2023-10-30

**Soundness:** 1 poor
**Presentation:** 2 fair
**Contribution:** 2 fair
**Rating:** 3
**Confidence:** 4

**Summary:**

The paper presents a self-supervised pre-training method for remote sensing pixel timeseries based on masked autoencoders. The pre-trained a Transformer backbone, which is either finetuned or used as a feature extractor in downstream tasks, is shown to have strong performance while having a small parameter count.

**Strengths:**

The paper deals with an important issue which is model pretraining using remote sensing data. For remote sensing applications of machine learning there exist plentiful unlabelled data directly from satellite while getting ground truths can be challenging, as a result labelled datasets for remote sensing are typically small and taking advantage of large scale unlabelled data through pretraining could bring significant benefits. As discussed, the masked autoencoder framework is by design suitable for dealing with missing data which is a desirable property for remote sensing applications.

**Weaknesses:**

I believe there are several issues with this paper that need to be addressed by the authors.

1) The application of Transformers to remote sensing should be discussed in more detail, e.g. [1] for pixel timeseries classification and [2] image timeseries classification and segmentation (despite operating on image timeseries base their model design on similar points as bullet points 1, 2 mentioned in the introduction).
[1] https://www.sciencedirect.com/science/article/pii/S0924271620301647
[2] https://openaccess.thecvf.com/content/CVPR2023/html/Tarasiou_ViTs_for_SITS_Vision_Transformers_for_Satellite_Image_Time_Series_CVPR_2023_paper.html

2) I believe the importance attributed to small model size is not as significant for the intended use case as it is presented in the paper. While it is always desirable to reduce model size and FLOPS as this reduces both latency, use of computing resources and their environmental footprint, models operating on satellite images will typically run on servers, and there is a limit to the number of inference runs imposed by the size of the area of interest. The revisit time of satellites (5 days for S2) provides a window for inference to run over large areas even with commodity hardware. This is in contrast to models operating on edge devices or models potentially receiving a very large number of calls on servers, e.g. chatbots, in which small model size can be crucial for deployment. If the authors have some specific use cases under consideration these should be presented and the argument should be made why a small model is needed. Also, one of the main findings of deep learning research, the fact the large overparameterized models outperform smaller ones even for small scale datasets, has not been successfully applied to satellite image timeseries in which performance typically drops after a certain model size. On that last part, an ablation in terms of model size would be interesting to showcase how the proposed framework scales.

3) While the authors recognise the need for methods particularly designed for remote sensing data, other than the domain specific inputs,  the proposed method itself does not account for such characteristics, utilising a pre existing framework (MAE) and architectures (Transformers) without significant modifications.

4) a) In 2.2 the details of tokenization are not clear. Different channel groups are separately encoded to a common latent space of size d_e but it is not clear if these encodings are concatenated channel wise (Fig.1 suggests so, in this case the encoding size of each group should be mentionned) or side by side (E \in R^{T |Cdyn| + |Cstatic| x d_e} suggests so). Also, the paper mentions the use of different learnable encodings for different channel groups, why are these needed if channels have fixed positions over the input? This part needs to be explained better for understanding.

5) Evaluation: a) Several deep learning works for remote sensing are presented in page 1 but these are not used as baselines in experiments despite being shown to outperform non deep learning baselines in original studies. As an absolute minimum, the Presto encoder architecture (Transformer) should be trained from random initialization on all downstream tasks to isolate the effect of pre-training. This is the most important point of critisism i have.
b) In Table 1 random timesteps appears to have practically the same performance as all transformations combined.
c) Section 3.2 discusses performance in image based tasks despite the model not containing mechanisms for spatial modelling, it basically operates as a deep fully connected network on single pixel features. It has to be clarified what can be derived from such a comparison. In Table 4 it appears that both baselines outperform Presto. In Table 5 experiments it is shown that Presto outperfroms baselines specifically trained on spatial data which is very surprising given that Presto has no understanding of spatial relations and that the MAE family of models are very similar in terms of their pre-training objective. This most probably can be attributed to the fact that there is a train/test input size discrepancy for the baselines. Given that both models operate out of domain not a lot can be derived from this comparison. If the argument is related with the need for reduced algorithmic complexity then this not a fair comparison for the baselines which could be trained at smaller resolutions and be better and less expensive to compute at the same time. Overall, I dont believe that reduced algorithmic complexity can justify the use of a non spatial model in these spatial modelling tasks (see point 2). e) Experimental results should include more performance metrics including pixel based and class based metrics. I propose to move tables from supplementary material to main paper. Also, several captions need to provide more details to make them complete for understanding (Table 1: class or pixel based metric, Tables 4, 5: what are top, bottom?).

**Questions:**

1) Will the pretraining data be shared as part of the submission to help replication?
2) How does a Transformer trained from scratch in downstream tasks compare with the pre-trained model?
3) Regarding Image based tasks, the reasons why a pixel-timeseries model outperfroms spatial-only models in Table 5 should be analysed. a) What is the baseline performance and FLOPS when trained at test resolution? and what is the performance using 224x224 inputs?
b) Do both baselines and Presto use the same input data or is Presto informed with additional details about location, elevation?
4) How does Presto scale with model size/pretrain-data/finetune-data size?

---

> ### Author Response · Authors · 2023-11-17
> **Response to Reviewer DF48 (part 1)**
>
> Thank you for your thoughtful feedback.
>
> - **Additional discussion of related works**: Thank you for highlighting these papers. We already cite [1] in our introduction, in the paragraph titled “A highly informative temporal dimension”. Based on your suggestion, we add a paragraph in our introduction discussing the use of transformers for remote sensing time-series.
>
> - **Importance of model size**: Our apologies for not being clearer in the paper. It is indeed surprising, but in practice, a very large number of remote sensing applications are bottlenecked by computation. We've seen this in our own collaborations with public agencies deploying remote sensing algorithms. Even in rich, well-resourced countries, governments often do not have the computational resources (yet) to deploy highly compute-intensive algorithms, and this is far worse in poorer countries. Most such applications are currently using less effective algorithms to save on computation. We highlight quotes below from practitioners which emphasise the importance of model size - not just for latency, but for total cost:
>   - "Becoming Good at AI for Good" [3] by authors from Microsoft, has a subsection titled “Model development with resource constraints”. We quote the following specific use case from this paper:
>
>         “ Robinson et al. [51] trained a fully convolutional neural network (CNN) on over 55 terabytes of aerial imagery to create a high-resolution land cover map over the United States. Differences in seconds of running time per batch translate to hundreds of dollars in the cost of the final computation. Here, a larger, state-of-the-art model would incur a ~270% increase in the cost of the final computation for a fractional increase in performance metrics such as accuracy and intersection-over-union, and so a trade-off in favor of lowering the cost was made.
>
>   - The European Space Agency’s World Cereal program, which mapped cropland at a global scale at a 10m resolution, shared a blog post of their experience (https://blog.vito.be/remotesensing/worldcereal-benchmarking). They emphasise three key criteria when selecting models:
>
>         “(i) Quality-related (e.g. how accurate are the results?), (ii) Performance-related (e.g. what computational resources are needed?), (iii) Requirements-related (e.g. how much and what kind of training data is required?)”
>
>   - The SoilsGrid project [4] mapped multiple soil metrics at a global scale, at a 250m resolution. We quote two areas from their paper where the discuss the computational costs, and modelling considerations as a result of these costs:
>
>         “The total size of all input and output data used to generate SoilGrids exceeds 30 TiB, so that a first step in preparing SoilGrids250m was to obtain a Synology 12-Bay NAS storage server with 60 TiB space. Handling such a large data set presented major challenges considering computational complexity and network bandwidth limitations.”
>
>       They omit kriging, which may increase performance, because of the computational costs:
>
>         “... overall kriging of residuals for global land mass does not seem to be necessary nor is it practical to implement for billions of pixels: it would only marginally improve the accuracy of predictions at high computing costs.”
>
>   - MOSAIKS [5] develops a feature-database from satellite imagery specifically with the goal of reducing the computational cost of applying machine learning algorithms to satellite imagery. We quote their introduction:
>
>         “The resource requirements for deploying SIML technologies, however, limit their accessibility and usage. Satellite-based measurements are particularly under-utilized in low-income contexts, where the technical capacity to implement SIML may be low, but where such measurements would likely convey the greatest benefit. For example, government agencies in low-income settings might want to understand local waterway pollution, illegal land uses, or mass migrations. SIML, however, remains largely out of reach to these and other potential users because current approaches require a major resource-intensive enterprise, involving a combination of task-specific domain knowledge, remote sensing and engineering expertise, access to imagery, customization and tuning of sophisticated machine learning architectures, and large computational resources.”
>
> [3] Kshirsagar, Meghana, et al. "Becoming good at AI for good." Proceedings of the 2021 AAAI/ACM Conference on AI, Ethics, and Society. 2021.
>
> [4] Hengl, Tomislav, et al. "SoilGrids250m: Global gridded soil information based on machine learning." PLoS one 12.2 (2017): e0169748.
>
> [5] Rolf, Esther, et al. "A generalizable and accessible approach to machine learning with global satellite imagery." Nature communications 12.1 (2021): 4392.

---

> > ### Author Response · Authors · 2023-11-17
> > **Response to Reviewer DF48 (part 2)**
> >
> > - **Adapting the method for satellite data**: We highlight the following ways in which our model architecture is adapted to satellite data:
> >   - **The use of location metadata as inputs**: While location metadata is always available in remote sensing datasets (e.g. EuroSat), it is rarely used by models. Our method leverages this location metadata.
> >   - **The use of temporal encodings**: We adapt the traditional positional encodings used by transformer models to additionally consider the month which a token comes from. We discuss this in Section 2.2.
> >   - **Channel aware encodings**: We additionally include a learnable encoding which communicates to the model which sensor a token came from. This critical detail allows us to include multiple sensor inputs into our model; we are the first MAE-based model to support multiple sensors.
> >
> >   In addition, we note the following modifications to the MAE pre-training approach:
> >   - **A masking strategy with downstream tasks in mind**: As described in Section 2.3, we introduce (and ablate) a masking strategy which is aware of the structure of the data. Missing channels is a unique characteristic of remote sensing data which this method is specifically designed to address.
> >   - **The introduction of categorical and continuous inputs**: We introduce the ability to ingest categorical and continuous inputs at pre-training (and downstream) time. This allows for information-dense inputs (e.g. landcover maps) to be used in pre-training, allowing for semantic density to be introduced during pre-training.
> >
> > - **Tokenization details**: Our apologies for the lack of clarity. There is one encoding per timestep per channel group, so T * C (plus one for each static variable). These encodings do not get concatenated channel-wise, they serve as distinct inputs to the transformer model, which is why they need a channel group embedding (or there would be no way for the transformer to distinguish e.g. RGB tokens from precipitation tokens, which would likely make learning harder). The dimension formula in section 2.2 is correct.
> >
> >   Fig. 1 could indeed be interpreted as concatenating encodings channel-wise, but that is not the case, we just used a 2D representation of the inputs with a dimension for channels and another dimension for time, for clarity. In practice, all the encodings (cubes in the figure) get flattened into a 1D sequence before being fed to the transformer. We clarified this in the caption.
> >
> > - **Evaluation**: Thank you for your thorough feedback with respect to our evaluation methods.
> >
> >   - **the Presto encoder architecture (Transformer) should be trained from random initialization on all downstream tasks to isolate the effect of pre-training**:
> >
> >     Thank you for this suggestion; we have added experiments comparing finetuning the pre-trained model versus finetuning from scratch. These results are available in full in Section A.4 of our updated manuscript.
> >
> >     We quote Section A.4 below:
> >
> >         “To understand the effect of pre-training Presto, we finetune Presto and train it from scratch on EuroSat (Table 7), the regression tasks (Table 5 in the main paper) and TreeSatAI (Table 11). We omit the CropHarvest dataset because it was expressly designed as a few-shot-learning dataset. Its small size makes the construction of validation sets with which to control the finetuning (e.g. with early stopping) challenging.
> >
> >         Overall, we find a consistent and significant improvement from the use of pre-trained Presto compared to a randomly initialized version of the model. For the EuroSat task, pre-training consistently delivers an incresse in accuracy score > 0.1 (representing increases in accuracy of up to 25%). This effect is consistent with what we observe on the TreeSatAI dataset (where pretraining consistently increases mAP by around 10 points) and on the regression tasks (where pretraining reduces RMSE by to 15% on the algae blooms task)."
> >
> >   - **In Table 1 random timesteps appears to have practically the same performance as all transformations combined.**
> >
> >     Yes - however, we note that (in addition to performance gains), an advantage of this pre-training method with combined masking is that it reflects regimes in which the model will be used for downstream tasks (e.g. with missing channels, or missing timesteps or combinations of both). We added a clarification in the text in section 2.3. We also renamed the "Random timesteps" masking strategy to just "Timesteps", to avoid confusion. The combined strategy is clearly better than the "Random" one, which is what we meant in the text.

---

> ### Author Response · Authors · 2023-11-17
> **Response to Reviewer DF48 (part 3)**
>
> - **Evaluation (continued)**
>   - **Section 3.2 discusses performance in image based tasks despite the model not containing mechanisms for spatial modelling, it basically operates as a deep fully connected network on single pixel features. It has to be clarified what can be derived from such a comparison.**
>
>     We agree that Presto is primarily designed for pixel-timeseries. Note that because Presto always gets a lat/lon token as input, in addition to other channel tokens (like the RGB one), there are always at least 2 tokens that can attend to each other in self-attention layers.
>
>     Our goals when benchmarking Presto on scene classification tasks are:
>     - **Comparing Presto to comparable self-supervised models.**: Since all globally pre-trained self-supervised models for remote sensing ingest image-based datasets, comparing Presto to works like Seasonal Contrastive Learning, SatMAE or ScaleMAE require us to evaluate Presto on image-based datasets.
>
>     - We show that even a pixel-based small, efficient model with a simple spatial aggregation scheme achieves good performance on some scene classification tasks. Rather than outperforming very large models pre-trained specifically for spatial modelling, we offer a new choice that can be optimal for different preferences w.r.t. the performance/efficiency trade-off.
>
>     - Demonstrating the **wide applicability of Presto to different types of tasks**. Users might be interested in one model that handles both pixel-based and image-based tasks, for ease of use.
>
>   - **In Table 4 it appears that both baselines outperform Presto.**
>
>     Yes - we discuss this in Section A.3.1, and hypothesize that this is in part due to Presto’s (currently) coarse approach to modelling spatial information.
>
>   - **In Table 5 experiments it is shown that Presto outperforms baselines specifically trained on spatial data which is very surprising given that Presto has no understanding of spatial relations and that the MAE family of models are very similar in terms of their pre-training objective. This most probably can be attributed to the fact that there is a train/test input size discrepancy for the baselines. Given that both models operate out of domain not a lot can be derived from this comparison.**
>
>     We begin by noting that while Presto approaches the performance of the other MAE models, the best performing model given a full-resolution input is the ScaleMAE model, which achieves an accuracy of 0.960. In terms of train/test discrepancy, we use the (widely used) splits provided by [6], as implemented in the TorchGeo library. These additionally reflect the splits available in the ScaleMAE codebase [[link](https://github.com/bair-climate-initiative/scale-mae/tree/main/mae/splits)], suggesting we used the same train/test splits as the other models.
>
>     If by input size you are referring to the resolutions: we note that the ScaleMAE includes the exact same experiment (Table 11), where they compare ScaleMAE and SatMAE performance on EuroSat for input resolutions 16, 32 and 64. In addition, the ScaleMAE model has been specifically designed to work for different input resolutions; their abstract states: "we present Scale-MAE, a pretraining method that explicitly learns relationships between data at different, known scales throughout the pretraining process." Hence, we do not agree that the baselines are operating out of domain, at least not to a greater extent than Presto (which has also been pre-trained just on 10m/pixel resolution data).
>
>   - **Experimental results should include more performance metrics including pixel based and class based metrics. I propose to move tables from supplementary material to main paper.**
>
>     Thank you for this feedback - we have been consistent with the metrics reported by (i) the datasets themselves, and (ii) works which benchmark against them. Which specific metrics would you additionally like to see reported?
>
>   - **Also, several captions need to provide more details to make them complete for understanding (Table 1: class or pixel based metric, Tables 4, 5: what are top, bottom?).**
>
>     Our apologies for the lack of clarity. Table 1 computes pixel-based metrics.
>
>     - In Table 4, the top and bottom represent different inputs (S1 and S2), as denoted by the “Data” column
>     - In Table 5, the top row of Presto represents results with RGB inputs, and the bottom with multispectral (MS) inputs.
>
>     We have updated the tables and the task descriptions in Section 3.2 to make this clearer.
>
>     Please let us know if we have misunderstood which top / bottom you were referring to.
>
> [6] Neumann, Maxim, et al. "In-domain representation learning for remote sensing." arXiv preprint arXiv:1911.06721 (2019).

---

> > ### Author Response · Authors · 2023-11-17
> > **Response to Reviewer DF48 (part 4)**
> >
> > - **Questions**
> >
> >   - **Will the pretraining data be shared as part of the submission to help replication?**
> >
> >     Yes, we will publish the pre-training data after publication. All scripts used to export it from Earth Engine are currently available in the anonymous repository (https://anonymous.4open.science/r/presto-C4AB).
> >
> >   - **How does a Transformer trained from scratch in downstream tasks compare with the pre-trained model?**
> >
> >     We answer this question in part 2 of our answer ([link](https://openreview.net/forum?id=Iip7rt9UL3&noteId=3hJMeoUJxM)) above.
> >
> >   - **Regarding Image based tasks, the reasons why a pixel-timeseries model outperfroms spatial-only models in Table 5 should be analysed. a) What is the baseline performance and FLOPS when trained at test resolution? and what is the performance using 224x224 inputs?**
> >
> >     We want to emphasize that both SatMAE and ScaleMAE use EuroSat as a benchmark in their paper. ScaleMAE uses another even lower resolution dataset, and has an experiment with ScaleMAE and SatMAE results for the downscaled (16, 32) versions of EuroSat. Their model is both trained and tested on 224x224 inputs, both in their paper and in all results that we include in our paper. We cannot for obvious reasons retrain SatMAE and ScaleMAE on different resolutions, and we do not think that that would be within the scope of our paper.
> >
> >     We also note that [7] specifies that inputs should be resized to the pre-training image size (in this case 224 x 224 for SatMAE and ScaleMAE) for best results.
> >
> >   - **Do both baselines and Presto use the same input data or is Presto informed with additional details about location, elevation?**
> >
> >     In the EuroSat experiments, Presto gets the same inputs as the baselines (RGB or Multispectral S2 data), plus the encoded lat/lon coordinates. No elevation, topography or other input channels are used.
> >
> >   - **How does Presto scale with model size/pretrain-data/finetune-data size ?**
> >
> >     Thank you for this suggestion. We add an experiment showing how Presto’s performance scales with model size; we find that as model size increases, Presto’s performance increases when measured against the CropHarvest validation set. The results are available in Table 8 and are discussed in Section A.6.
> >
> > [7] Corley, I., Robinson, C., Dodhia, R., Ferres, J. M. L., & Najafirad, P. (2023). Revisiting pre-trained remote sensing model benchmarks: resizing and normalization matters. arXiv preprint arXiv:2305.13456.

---

> > > ### Comment · Reviewer_DF48 · 2023-11-22
> > >
> > > Dear Authors,
> > >
> > > I appreciate your comprehensive response to my initial comments.
> > >
> > > Your arguments have shifted my perspective on the role of model size in the tasks at hand, emphasizing the value of efficiency. I suggest incorporating some of these points into the paper's introduction to enhance the rationale for prioritizing the development of fast and efficient models.
> > >
> > > However, I remain unconvinced about the experimental validation process and the novelty of the method. The model, being exclusively temporal, lacks mechanisms for spatial modeling. Without concrete evidence demonstrating that the spatial modeling performance is specifically due to the proposed method, it's not clear that any powerful enough pixel timeseries model would behave differently. I recommend comparing the proposed method with similar models and pretraining techniques specific to pixel timeseries from existing literature. This comparison is crucial to understand the method's unique contributions. While the assertion that this class of models could substitute spatial models in compute-limited settings has some merit, it should not be a primary point of comparison.
> > >
> > > Regarding novelty, aspects like the handling of features from different sensors and the use of position encodings are interesting (rebuttal helped make this clearer for me). Yet, detailed ablations and analysis are necessary to ascertain their impact in contrast to standard architectures.
> > >
> > > In light of these considerations, I am maintaining my initial score. My advice for further strengthening your paper includes adding more baseline comparisons for both the pretraining approach and the model and to conduct thorough ablations on all newly introduced components to more clearly demonstrate their effectiveness.

---

> ### Author Response · Authors · 2023-11-22
> **Response to Reviewer DF48's official comment**
>
> Thank you for your thoughtful response.
>
> 1. **Comparing the proposed method with similar models and pretraining techniques specific to pixel timeseries from existing literature.**
>
>     While we agree that comparing the model to existing pre-trained pixel time-series models would be useful, no globally pre-trained models exist. We discuss this in Section 1:
>
>     > While several pre-training methods have been proposed for transformers with remote sensing timeseries [1,2,3], these have not aimed at multi-task, global applicability, having been pre-trained and evaluated on highly local areas (e.g., central California) and evaluated only for a single task (e.g., crop type classification).
>
>     In order to comprehensively evaluate Presto, we therefore:
>
>     - Compare Presto to the state of the art model for the task at hand, where possible (e.g. [4,5]).
>     - Compare Presto to pre-trained remote sensing models which treat remote sensing data as imagery [5,6,7] (requiring us to run Presto on the EuroSat dataset).
>
> 2. **Yet, detailed ablations and analysis are necessary to ascertain their impact in contrast to standard architectures.**
>
>     We completely agree that ablations are always helpful to better understand how design choices affect a model’s performance. We have included the following ablations for the model:
>     - (at your suggestion) ablations on the effect of pre-training, in Tables 5, 7 and 12
>     - (at your suggestion) ablations on model size in Table 8
>     - ablations on the pre-training masking strategies, in Table 1
>
>     We note that further ablations on model design choices would be tricky in our case, since the model would not function without them (e.g. without the channel encodings, the decoder would not know what to reconstruct).
>
> [1] Yuan Yuan and Lei Lin. Self-supervised pretraining of transformers for satellite image time series classification. IEEE Journal of Selected Topics in Applied Earth Observations and Remote Sensing, 14:474–487, 2020
>
> [2] Yuan Yuan, Lei Lin, Qingshan Liu, Renlong Hang, and Zeng-Guang Zhou. Sits-former: A pretrained spatio-spectral-temporal representation model for sentinel-2 time series classification. International Journal of Applied Earth Observation and Geoinformation, 106:102651, 2022.
>
> [3] Yuan Yuan, Lei Lin, Zeng-Guang Zhou, Houjun Jiang, and Qingshan Liu. Bridging optical and sar satellite image time series via contrastive feature extraction for crop classification. ISPRS Journal of Photogrammetry and Remote Sensing, 195:222–232, 2023.
>
> [4] Gabriel Tseng, Hannah Kerner, and David Rolnick. TIML: Task-informed meta-learning for crop type mapping. In AI for Agriculture and Food Systems at AAAI, 2022.
>
> [5] Steve Ahlswede, Christian Schulz, Christiano Gava, Patrick Helber, Benjamin Bischke, Michael Forster, Florencia Arias, Jorn Hees, Begum Demir, and Birgit Kleinschmit. Treesatai benchmark archive: A multi-sensor, multi-label dataset for tree species classification in remote sensing. Earth System Science Data, 2023.
>
> [6] Yezhen Cong, Samar Khanna, Chenlin Meng, Patrick Liu, Erik Rozi, Yutong He, Marshall Burke, David B. Lobell, and Stefano Ermon. SatMAE: Pre-training transformers for temporal and multi-spectral satellite imagery. In Alice H. Oh, Alekh Agarwal, Danielle Belgrave, and Kyunghyun Cho (eds.), NeurIPS, 2022
>
> [7] Colorado J Reed, Ritwik Gupta, Shufan Li, Sarah Brockman, Christopher Funk, Brian Clipp, Salvatore Candido, Matt Uyttendaele, and Trevor Darrell. Scale-mae: A scale-aware masked autoencoder for multiscale geospatial representation learning. arXiv preprint arXiv:2212.14532, 2022.
>
> [8] Oscar Manas, Alexandre Lacoste, Xavier Gir´o-i Nieto, David Vazquez, and Pau Rodriguez. Seasonal contrast: Unsupervised pre-training from uncurated remote sensing data. In CVPR, 2021.

---

### Official Review · Reviewer_5BXx · 2023-11-01

**Soundness:** 3 good
**Presentation:** 4 excellent
**Contribution:** 3 good
**Rating:** 6
**Confidence:** 4

**Summary:**

In this article authors proposed Pretrained Remote Sensing Transformer (Presto) a transformer architecture pretrained with remote sensing timeseries imagery. This model can generate pixel-wise feature representation from timeseries imagery from different remote sensing imagery sensors. Presto is pre-trained using a proposed self-supervised objective that leverages the structure and consistency of multi-sensor data. The article argues that Presto achieve competitive or superior performance on multiple geospatial ml downstream tasks like land cover mapping, crop type identification, and fuel moisture estimation, compared to larger models trained from scratch. Authors also comapre performance with other existing self-supervised methods.

**Strengths:**

* The article is well written, techincally sound, and easy to follow.
* **Reproducibility**. Code have been made available and it is easy to reproduce results. This is also very important to motivate adoption from the community.
* **Generizability**. Presto works well for multiple application tasks and was tested in multiple geographic locations.

**Weaknesses:**

* **Resolution constraints**. Most downstream tasks Presto is tested on use coarse resolution. It would be nice to test the effect of encoding different resolution imagery during pretraining
* **Small performance gains**. The performance improvement for some of the tasks tested compared to simple baselines is limited.
* **Set of downstream tasks tested**. It is worrisome to me that most self-supervised methods test their proposed approach againg a different set of downstream task maiking it hard to actually compare apporaches. There are multiple other self-supervised approaches that could be used as comparison.

**Questions:**

1. Do you have intuition on why the improvement from Presto on TreeSatAi is much larger on S1 than S2 on Teble 4.
2. Figure 2 would benefit by explaining which channel are missing which channels were masked.

---

> ### Author Response · Authors · 2023-11-17
> **Response to Reviewer 5BXx**
>
> Thank you for your review. We address specific points below:
>
> - **Resolution constraints**: We agree that Presto is only tested at the resolution of freely available satellite imagery (and highlight this as a limitation and area of future work in Sections 5 and A.3.1 of the paper). We note that the EuroSat runs are run at a variety of (decreasing) image resolutions - Presto’s performance remains relatively consistent as image resolution decreases.
>
> - **Small performance gains**: We emphasise that in many cases, we are benchmarking against state-of-the-art methods for a task. The main contribution of Presto is not to achieve state-of-the-art results on some benchmarks; it is instead to introduce a computationally efficient model which can be applied to a large range of remote-sensing data formats. We emphasise that no other pre-trained model would be capable of running on all of these benchmarks, let alone achieving competitive results on them.
>
>     In addition, we do note that performance gains are considerable for CropHarvest (Table 3), and for the EuroSat tasks (Figure 3) (especially when considering the compute vs. performance trade offs).
>
> - **Set of downstream tasks tested**: Thank you for this feedback. We have added other models to the EuroSat benchmark, to better illustrate the performance of Presto relative to other self-supervised models and model architectures. Specifically, we have added the Seasonal Contrast (SeCo) and GASSL papers as additional benchmarks for Presto.
>
>     We note that other self-supervised methods **cannot ingest** the full set of downstream tasks we test against (for example due to their inability to ingest the full timeseries or all the sensor inputs). This ability to process diverse modalities of data (which we fully describe in Table 2) is one of Presto’s key strengths.
>
> - **Intuition on TreeSat performance**: Our hypothesis is that the information gain from the implicit reconstruction (i.e. from Presto reconstructing S2, ERA5, Dynamic World and topography given S1) is much higher for S1 than from S2.
>
> - **Figure 2 caption**: Apologies for the confusion; in our caption, we state that “In this experiment, we mask the Sentinel-2 RGB channels". There are no other channels masked nor missing - we have updated the caption to specify this. Please let us know if there are any additional changes you would like to see to make this clearer.

---

### Author Response · Authors · 2023-11-17
**Summary of changes**

We thank all reviewers for their reviews, summarize the overall changes to our paper below. In the updated pdf, these changes are in blue for ease of identification.

Major changes

- As suggested by reviewers **5BXx** and **X32P**, we include comparisons to additional self-supervised methods in Figure 3.a. and Tables 6 and 7.
- As suggested by reviewer **DF48**, we include analysis to disentangle the effect of pre-training from the model architecture. These results are in new tables (Table 7) and added to existing tables (Table 11). In addition, we discuss these results in Section A.4.
- As suggested by reviewer **DF48**, we include experiments showing how Presto performance scales with model size (Table 8). We discuss these results in Section A.6.

Minor changes
- As suggested by reviewer **DF48**, we add a discussion about transformers for remote sensing time series in Section 1.
- As suggested by reviewer **5BXx**, we update the caption of Figure 2 to clarify what is masked and unmasked.
- To respect the 9 page limit, we move the baselines (section 3.3) into the appendix.

---

### Meta-Review · Area_Chair_MtRW · 2023-12-11

**Metareview:**

This paper develops a transformer-based model tailored to and pre-trained on remote sensing pixel time-series data from multiple complementary sensors.  Performance gains over larger models trained from scratch are demonstrated on several downstream tasks (via transfer learning) such as land cover mapping, crop type identification, and fuel moisture estimation.

The strengths of this paper are topic of great interests in remote sensing, clear motivation, computational efficiency, and reproducibility with provided code.   The weaknesses of the paper are unconvincing experimental validation, limited technical novelty, lack of analytical insights (into, e.g., how a model trained on pixel-times series can perform well in a spatial scene task), and unclear performance when the pixel resolution increases.

This paper has received 4 reviews and active rebuttals/discussions.  While the results are extensive (including both the initial/updated submission and the rebuttals), reviewers remain unsatisfied or lukewarm towards acceptance, based on which the AC recommends rejection.

The authors' great rebuttal efforts are very much appreciated!  The paper clearly has its values in remote sensing; the authors are encouraged to take all the comments and discussions into revision for another venue that would need little rebuttal.

**Justification For Why Not Higher Score:**

Reviewers remain unsatisfied or lukewarm towards acceptance, due to unconvincing experimental validation and limited technical novelty/insights.

**Justification For Why Not Lower Score:**

N/A

---

### Decision · Program_Chairs · 2024-01-16

Reject